EMBO
Molecular Medicine

# A novel genome-wide *in vivo* screen for metastatic suppressors in human colon cancer identifies the positive WNT-TCF pathway modulators TMED3 and SOX12

Arnaud Duquet, Alice Melotti, Sonakshi Mishra, Monica Malerba, Chandan Seth, Arwen Conod & Ariel Ruiz i Altaba*

## Abstract

The progression of tumors to the metastatic state involves the loss of metastatic suppressor functions. Finding these, however, is difficult as *in vitro* assays do not fully predict metastatic behavior, and the majority of studies have used cloned cell lines, which do not reflect primary tumor heterogeneity. Here, we have designed a novel genome-wide screen to identify metastatic suppressors using primary human tumor cells in mice, which allows saturation screens. Using this unbiased approach, we have tested the hypothesis that endogenous colon cancer metastatic suppressors affect WNT-TCF signaling. Our screen has identified two novel metastatic suppressors: TMED3 and SOX12, the knockdown of which increases metastatic growth after direct seeding. Moreover, both modify the type of self-renewing spheroids, but only knockdown of TMED3 also induces spheroid cell spreading and lung metastases from a subcutaneous xenograft. Importantly, whereas TMED3 and SOX12 belong to different families involved in protein secretion and transcriptional regulation, both promote endogenous WNT-TCF activity. Treatments for advanced or metastatic colon cancer may thus not benefit from WNT blockers, and these may promote a worse outcome.

**Keywords** cancer; *in vivo* assay; metastatic suppressor; WNT-TCF
**Subject Categories** Cancer; Ageing

## Introduction

Nearly one-half of patients with advanced local colon cancer will develop metastasis in the liver and these carry a life expectancy of less than 1 year (Rothbarth & van de Velde, 2005; Gallagher & Kemeny, 2010). Understanding how metastases are normally suppressed could open new therapeutic avenues. A number of metastatic suppressors have been identified and these include membrane, nuclear and cytoplasmic factors with highly divergent functions, such as mitogenic, survival, RNA modulation and cell-cell signaling regulation (Rinker-Schaeffer *et al*, 2006; Mehlen & Puisieux, 2006; Pencheva & Tavazoie, 2013). Such variety likely reflects the multiple strategies metastasizing cancer cells use, but the heterogeneity makes the choice of components to target uncertain. In addition, defining metastatic suppressor functions remains difficult, and the heterogeneity mentioned above could derive from limited *in vitro* assays and the use of clonal cell lines, neither of which recapitulates *in vivo* metastases. A more comprehensive view of metastatic suppressors is thus urgently necessary, and this must use *in vivo* selection of metastatic cells and relevant primary human cancer cells that have not been cloned.

In contrast to metastatic suppressors, targeting common tumor-initiating oncogenic events has generated enthusiasm since these are thought to be required at all stages of tumor progression, including metastasis. The common tumor-initiating event in human colon cancers is hyperactive WNT-TCF signaling, usually through mutation of the negative regulator adenomatous polyposis coli (APC) and the consequent constitutive activation of β-CATENIN, which then interacts with nuclear TCF/LEF factors to regulate target gene transcription (Morin *et al*, 1997; MacDonald *et al*, 2009; Valenta *et al*, 2012).

The idea that WNT-TCF signaling is essential from the formation of benign adenomas to the expansion of lethal metastases (Kinzler & Vogelstein, 1996) has led to great efforts to block WNT-TCF function with small molecules for therapeutic purposes, aiming to eradicate so far incurable metastases (Anastas & Moon, 2013). This idea is based on three key results: (i) The majority of human colorectal cancers harbor deletions in *APC* (Kinzler & Vogelstein, 1996); (ii) a dominant-negative form of TCF (dnTCF), which blocks all TCF/LEF function, inhibits the proliferation of early and advanced human colon cancer cells *in vitro* (van de Wetering *et al*, 2002); and (iii) constitutive activation of the canonical Wnt-Tcf pathway by loss of *Apc* induces intestinal adenomas in mice (Su *et al*, 1992).

The requirement of WNT-TCF in advanced colon adenocarcinomas and metastases, however, has been recently challenged (Varnat

Department of Genetic Medicine and Development, University of Geneva Medical School, Geneva, Switzerland
*Corresponding author. Tel: +41 22 379 5646; E-mail: Ariel.RuizAltaba@unige.ch

 

*et al*, 2010) based on a number of findings: (i) The levels of expression of WNT-TCF target genes are downregulated in advanced colon cancers of patients with metastases and in liver metastases, as compared with local tumors without metastases; (ii) *in vitro* culture imposes WNT-TCF–dependency to advanced colon adenocarcinomas and metastases; and (iii) WNT-TCF pathway blockade by dnTCF in human colon cancer cells enhanced metastatic growth in the lungs of host mice after direct cell injection into the circulation. However, it remained unclear whether endogenous metastatic suppressors would affect, directly or indirectly, WNT-TCF function.

To address these issues and provide a general view of metastatic suppressor functions in primary cancer cells, we have designed a novel genome-wide screen strategy *in vivo*: Marked human cancer cells with specific genetic alterations (e.g. expressing shRNAs) are selected for enhanced metastatic behavior in mice. Importantly, screens can reach saturation in independent hosts, thus providing a comprehensive view.

We have implemented this novel and unbiased assay to provide the proof of principle of its usefulness with one primary colon cancer cell population. Moreover, this allowed us to test the hypothesis derived from our previous work (Varnat *et al*, 2010) that at least some endogenous colon cancer metastatic suppressors should affect WNT-TCF signaling, as assayed by the determination of pathway-specific TCF target gene signatures. Here, we report the identification of two genes with novel *in vivo* metastatic suppressor functions: *TMED3* and *SOX12*. These two very different proteins share a common function in supporting endogenous TCF target responses, but may function at opposite ends of the WNT-TCF signaling cascade. TMED3 is a member of a family of p24 proteins previously involved in WNT ligand secretion through the endoplasmic reticulum (ER)-Golgi system (Buechling *et al*, 2011; Port *et al*, 2011). SOX12 is a nuclear transcription factor and member of the SOXC family that includes SOX11 and SOX4, which regulates β-CATENIN/TCF transcriptional output (Sinner *et al*, 2007). Whereas both affect the type of spheroids formed in clonogenic assays *in vitro* and metastases *in vivo* after direct injection into the circulation, only interference with the function of TMED3 induced full metastasis from a primary subcutaneous xenograft to a distant organ. Our data validate a novel assay for metastatic suppressors *in vivo* using primary human cancer cells and identify two novel metastatic suppressors that are positive WNT pathway regulators. Together with previous work (Varnat *et al*, 2010), the results suggest that, paradoxically, endogenous WNT signaling simultaneously promotes primary tumorigenesis and prevents metastasis.

# Results

## A novel genome-wide *in vivo* screen for metastasis suppressors with primary human colon cancer cells

To perform the screen in a primary human colon cancer, we have used early passage primary CC14 Tumor-Node-Metastasis 4 (TNM4) human colon cancer cells (Varnat *et al*, 2009) since these maintain the original morphology and form characteristic epithelial islets *in vitro* and crypt-like tumor structures *in vivo* in xenografts (Fig 1A; Varnat *et al*, 2009, 2010). CC14 cells belong to the transit amplifying/inflammatory type (Supplementary Fig S1) according to a recent classification scheme (Sadanandam *et al*, 2013).

CC14 cells were transduced with a *lacZ*[+] lentivector to allow the visualization of βGal[+] cells (Stecca *et al*, 2007). $2 \times 10^6$ CC14[lacZ+] cells were additionally transduced with a genome-wide lentiviral shRNA library or with the empty pSIH-GFP[+] vector used for library construction as control at a multiplicity of infection of 1–2 (Fig 1A), representing 10-fold the library size of $2 \times 10^5$ independent shRNAs. After 4-fold expansion, $1 \times 10^6$ library-expressing or control cells were injected into each of eight nude mice intravenously so that every injected mouse received cells with 5-fold coverage of the shRNA library. Intravenous injection leads to the efficient seeding of the lungs within 15–60 min (Mendoza *et al*, 2010; data not shown; see Materials and Methods).

Eight weeks after tail vein injection, the lungs of host mice were dissected and one right lobe of each mouse was stained for βGal[+] metastasis visualization. 5 of 8 mice showed increased number of βGal[+] metastases (Fig 1C–E) over controls (Fig 1B). The lobes were each cut into 5–10 (for each right lobe) or 20 (for the left lobe) slices, and the DNA of each slice amplified by PCR to detect and clone shRNA inserts present in metastatic tumor cells. Sequencing revealed that about half of the slices contained metastatic human cells with 1–5 independent shRNAs per PCR[+] slice (Fig 1F), indicating a high incidence of independent metastatic sites.

Two shRNA sequences targeting *TMED3* and *SOX12* were repeatedly found in 4 of 5 animals with increased βGal[+] metastases and in 11 of 49 and 5 of 49 PCR[+] slices, respectively (Fig 1F), indicating that the screen was near saturation.

The top 7 candidates (Fig 1F) were first tested for target mRNA expression. Three of them (*CD163, HIC2* and *KERATIN6B*) were not detected in CC14 cells, suggesting off-target effects of the respective shRNAs. The remaining four were retested individually in the original pSIH-GFP[+] lentivector (Fig 2A and B), and all yielded increased metastatic colonies over controls.

Secondary analyses with an independent shRNA in a different lentivector backbone (pGIPZ-GFP[+]), with targeting efficiencies of 60–90% (Supplementary Fig S1), confirmed the metastatic suppressor activity of only two genes: *TMED3* and *SOX12* (Fig 2C and D), which were also the top hits of the screen (Fig 1F). These two genes were found to be widely expressed in various primary and colon cancer cell lines (Supplementary Fig S1).

The shRNA targeting *TEAD1*, which encodes a mediator of the HIPPO-YAP pathway (e.g. Zhao *et al*, 2011), was the second most abundant shRNA (4.4% of all sequences vs. 6.6% for *shTMED3* and 2% for *shSOX12*; Fig 1F). Even though the second shRNA against *TEAD1* did not yield more metastases, we sought to further address the role of this pathway by targeting YAP1 itself, given that there are multiple TEAD family members with overlapping functions (Sawada *et al*, 2008; Zhao *et al*, 2011) and, importantly, that TEAD-YAP1 function has been previously implicated in colon cancer (Lamar *et al*, 2012; Chen *et al*, 2012; Rosenbluh *et al*, 2012). Tail vein injection of cells expressing *shYAP1* did not increase the number of metastases, supporting the second *shTEAD1* result. Since CC14 cells alone form few lung colonies, we sought to use fully immunocompromised NSG mice in which these cells form more metastases to test whether YAP1 inhibition could in fact have the opposite effect. Injection of viable CC14-*shYAP1* cells into the circulation of NSG mice resulted in the near complete loss of metastatic colonies (Fig 2E, Supplementary Fig S2), supporting a

                                                                    

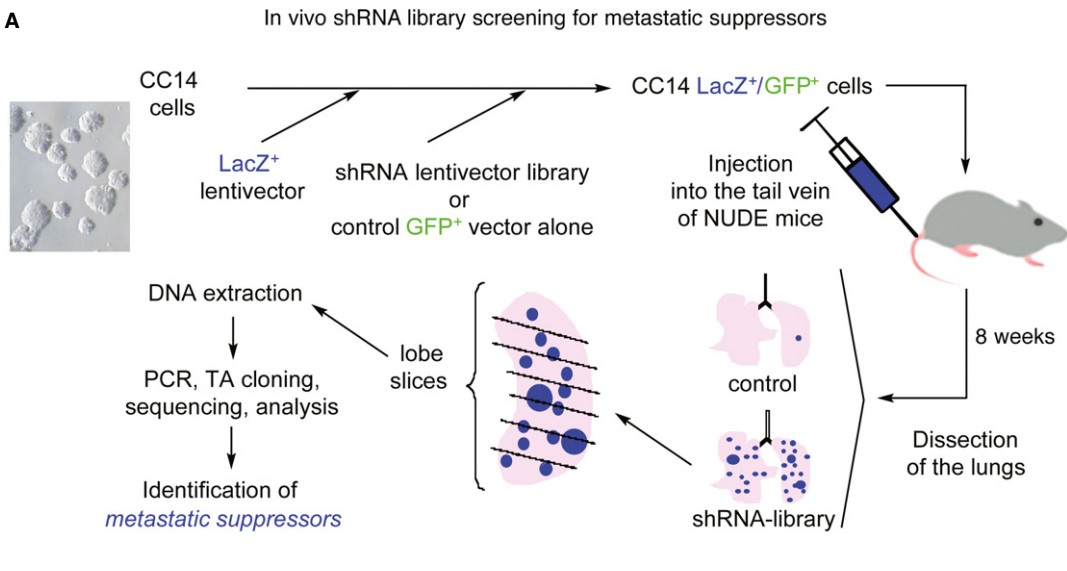

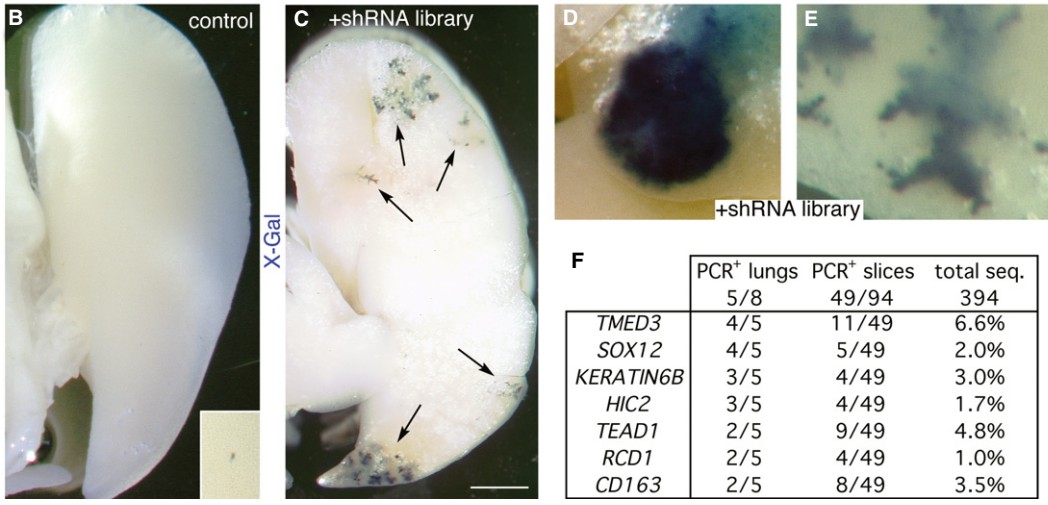

**Figure 1. *In vivo* screening for metastatic suppressors in CC14 colon cancer cells.**

A   Scheme representing the *in vivo* shRNA library screening approach in nude mice leading to the identification of pro-metastatic shRNAs as described in the text.

B, C   Representative images of left lung lobes from mice injected with control CC14 (B) or shRNA library-infected CC14 (C) cells after dissection and staining with X-Gal to reveal *LacZ*[+] human colon cancer metastatic cells. Inset in (B) shows a small and rare metastasis in the control condition at high magnification. Arrows (C) indicate stained metastases. Scale bar = 2 mm.

D, E   High-magnification images of lung metastases from mice injected with shRNA library-infected CC14 cells. Scale bar = 400 μm.

F   Summary of the quantification of PCR[+] lung, slices and total sequences per candidate gene.

pro-metastatic role of this pathway, behaving in the opposite manner to *shTMED3* and *shSOX12*.

### Knockdown of SOX12 and TMED3 enhance metastases in HT29 colon cancer cells

To extend the findings in CC14 cells with TMED3 and SOX12 presented above, we used HT29 human colon cancer cells to test the effects of *shTMED3* and *shSOX12* (Fig 3). Both reduced expression of the respective targets by 80–90% *in vitro* (data not shown) and enhanced metastases *in vivo* (Fig 3B) with variable penetrance. In this context, SOX12 KD yielded the strongest phenotype (Fig 3B and C).

### Interference with TMED3 or SOX12 confers a competitive advantage *in vivo*

The ability of knockdown of TMED3 and SOX12 to increase metastatic seeding and size could result from enhanced cell proliferation. BrdU incorporation analyses did not reveal differences between *shTMED3*, *shSOX12* and control cells (Supplementary Fig S3). Similarly, no growth enhancement was detected in subcutaneous tumors in nude mice composed of 100% *shTMED3*-GFP[+]-sorted or 100% *shSOX12*-GFP[+]-sorted cells as compared with controls (Supplementary Fig S3).

However, *in vivo* red/green competition assays (Fig 4A; Varnat *et al*, 2009; Zbinden *et al*, 2010) showed that both *shTMED3* and

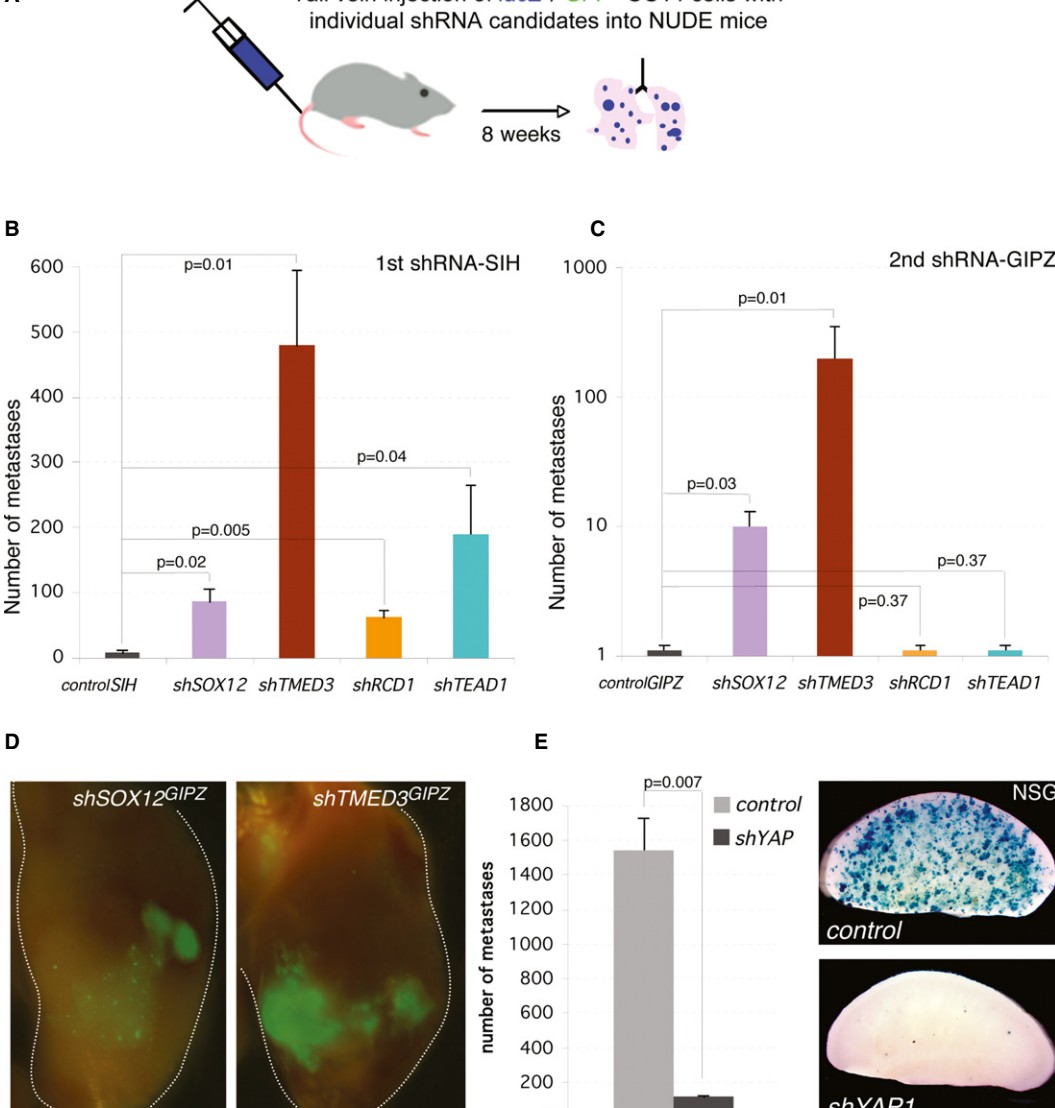

**Figure 2.** *In vivo* validation of metastastic suppressor candidates.

A    Scheme of tail vein injections into nude mice used to validate individual targets.

B, C  Histograms of the number of metastases obtained after injection of cells carrying different candidate shRNAs as noted. Two different shRNAs in different backbones were used for each target: SIH (B) and GIPZ (C).

D    GFP fluorescence images of a whole left lung lobe from *shSOX12* and *shTMED3*-injected mice. The lobe contour is denoted by a white line. Scale bar = 2.5 mm.

E    Histogram of the quantification of *LacZ*+ metastases (left) and images of representative left lobes after X-Gal staining (right) for control cells and cells carrying an shRNA targeting *YAP1*. Scale bar = 4 mm.

Data information: (B, C, E) *n* = 3 mice per condition. Error bars = ±s.e.m.

---

*shSOX12* endowed GFP+-expressing cells with an advantage as compared with sibling CC14 cells transduced with a lentivector expressing RFP in subcutaneous grafts (Fig 4B). This advantage was enhanced upon *in vivo* passage and could be seen in whole tumor imaging (where overall tumor size did not vary; Fig 4B) and in FACS analyses of dissociated tumor cells (Fig 4C and D). Red/green assays with *shYAP1*-GFP+ CC14 cells injected subcutaneously into nude mice revealed that YAP1 is cell autonomously essential for tumor growth as green cells disappeared from the tumor (Fig 4E

and F, Supplementary Fig S4), again showing the opposite phenotype of *shTMED3*- and *shSOX12*-expressing cells (Fig 4B and D).

## Knockdown of *TMED3*, but not *SOX12*, induces full metastasis to distant organs *in vivo*

To test whether knockdown of *TMED3* or *SOX12* could promote full metastasis from a primary tumor to a distant organ, we subcutaneously grafted CC14 *lacZ*+ cells expressing pSIH lentivectors into

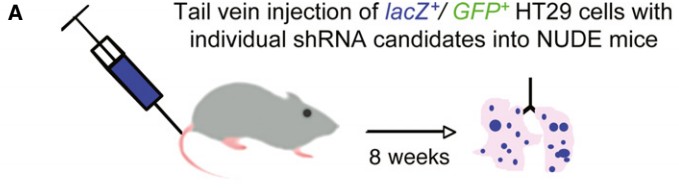

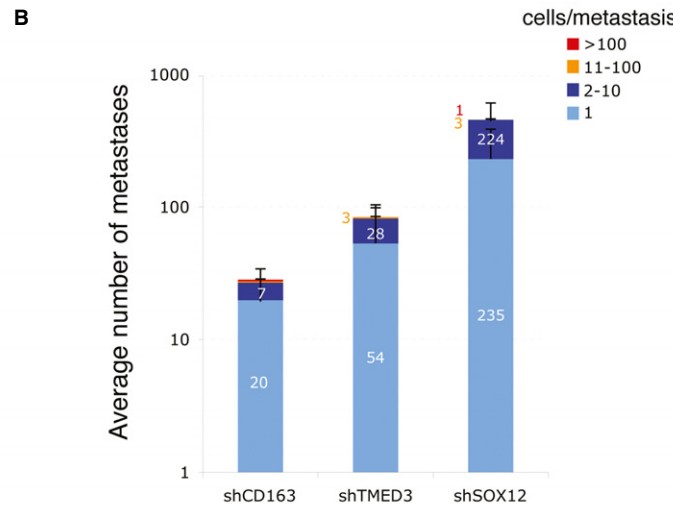

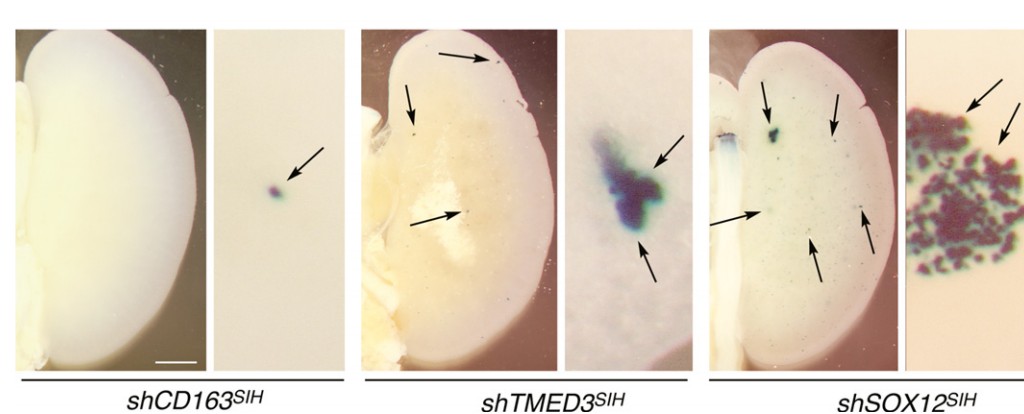

**Figure 3. *shSOX12* and *shTMED3* enhance metastases from HT29 colon cancer cells.**

A  Scheme of tail vein injections into nude mice.

B  Quantification of the number of metastases by metastasis size n control (shCD163) cells and in cells expressing *shTMED3* or *shSOX12* in the SIH lentivector backbone as indicated.

C  Representative images of X-Gal-stained left lungs (left panels) and details of metastases (right panels) from mice harboring cells as indicated. Scale bar = 2 mm (left panels), 400 µM (right panels).

nude mice and scored for the presence of βgal $^+$ colonies in the lungs (Fig 5A). Control cells yielded few lung βgal $^+$ colonies and the same result was obtained with *shSOX12* or *shKERATIN6B* tumors (Fig 5B and C), the latter used as an additional control (see above, Fig 1F). *shTMED3* subcutaneous tumors, in contrast, yielded 3- to 6-fold more and larger metastases using SIH or GIPZ backbones (Fig 5B, D and H; Supplementary Fig S5). Importantly, this was not due to an increase in tumor weight (Fig 5C and E).

To extend the result that *shTMED3* enhances full metastases, we used NSG mice, which are more permissive as they are fully

immunocompromised (Fig 5F). Subcutaneous engraftment of *shTMED3*-expressing cells into NSG mice resulted in the formation of more and larger lung metastases as compared with control lentivector-transduced cells (Fig 5F, G and I, Supplementary Fig S5).

**Both TMED3 and SOX12 promote large colon cancer stem cell sphere formation, but only TMED3 promotes sphere compaction**

To begin to delineate the function of TMED3 and SOX12 in colon cancer stem cells, which may underlie metastases (e.g. Sampieri &

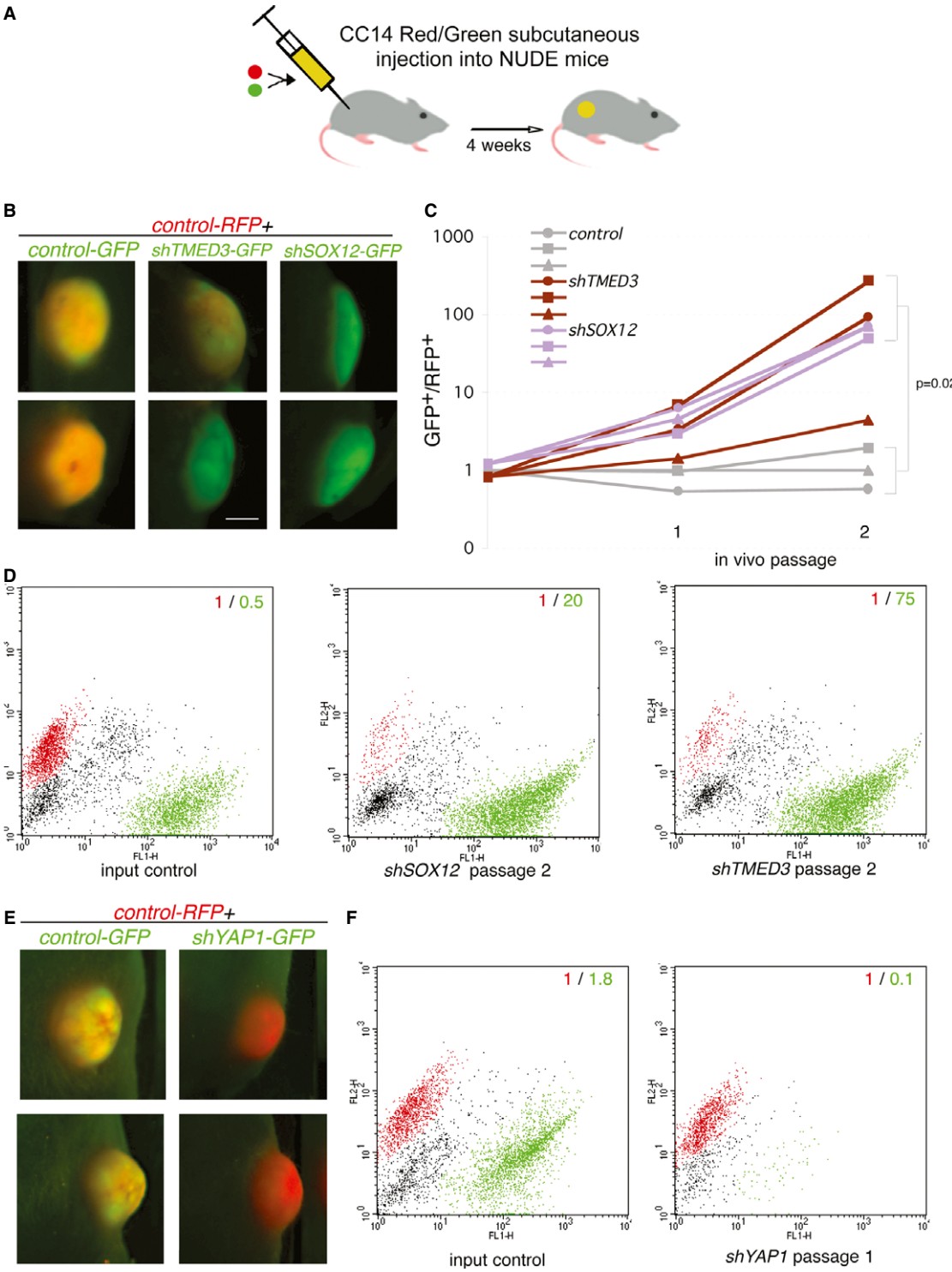

**Figure 4. *In vivo* Red/Green competition assays.**

A    Scheme of the *in vivo* red/green competition assay used to test the effect of individual shRNAs (see Varnat *et al*, 2009; Zbinden *et al*, 2010). Here, lentivector-transduced experimental GFP+ green and control transduced RFP+ red cells are mixed in equal amounts and co-injected *in vivo*.

B–F    Fluorescent images of two (top and bottom rows in each case) representative subcutaneous tumors for each condition after one (E) or two (B) *in vivo* passages, with shRNAs as noted in the figure. Green color indicates a predominance of GFP signal, whereas red color indicates a predominance of RFP signal. In control conditions with an empty GFP+ lentivector, the tumors retain similar amounts of RFP+ and GFP+ cells and the merged signals give a yellow color. (C) Quantification of the GFP+/RFP+ cell ratio, as determined by FACS analyses after cell dissociation, of control, *shTMED3* and *shSOX12* tumors over 2 passages *in vivo* as noted. Representative FACS plots of red and green populations for the given conditions. Note the opposite behavior of *shSOX12* and *shTMED3* (D) as compared with *shYAP1* (F; Supplementary Fig S4). Scale bar = 3.5 mm.

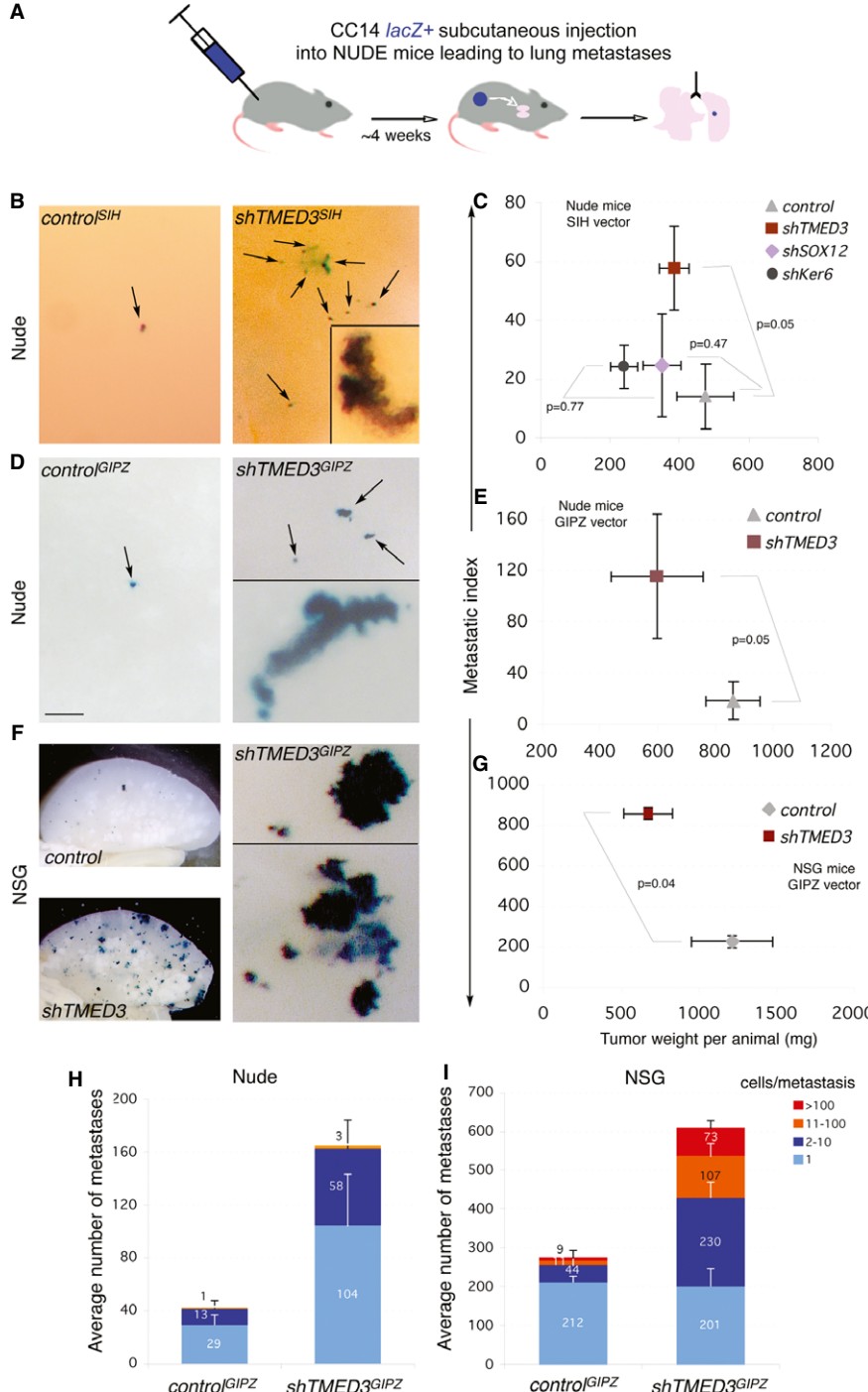

**Figure 5.  Induction of full metastasis *in vivo* by interference with TMED3.**

A    Scheme of the subcutaneous-to-lung metastatic assay *in vivo* used with CC14 cells in nude or NSG mice.

B–G  Images of left lobe lungs from nude (B, D) or NSG (F) mice after X-Gal staining at different magnification. Images in (B) and (F right panels) show high magnifications. Arrows indicate *LacZ*[+] metastases detected after the X-Gal reaction. Insets show additional metastases at high magnification. Histograms of the number of metastases detected in three independent experiments with the shRNAs in the SIH lentivector backbone in nude mice (C) or in the GIPZ lentivector backbone either in nude (E) or in NSG (G) mice. The *x*-axes represent the total tumor weight carried per animal in mg. The *y*-axes represent the metastatic index which is the number of metastases divided by the tumor weight per animal. The metastasis index incorporates the variable of tumor size as larger tumors tend to give more metastases in general. The graphs of metastasis index plotted against tumor weight thus allow the easy visualization of tumor behavior as a given cell type gives a characteristic metastatic index along tumor weight in a given animal model. Number of mice was for (C): 8 for control, 11 for *shTMED3*, 10 for *shSOX12*, 4 for *shKERATIN6B* (shKer); for (E): 6 for control, 6 for *shTMED3*; for (G): 4 for control, 4 for *shTMED3*. Scale bar = 250 µm (B, D, F right panel), 100 µM (insets in B, D), 3.5 mm (F left panels).

H, I  Quantification of metastases counted in the experiments shown in (E) and (G), respectively, ranked by the size of metastases as noted. Error bars = ±s.e.m.

Fodde, 2012), we first analyzed the effects of their knockdown in anchorage-independent clonogenic colonospheres, or spheroids, which report the ability and number of colon cancer stem cells to self-renew and form such clones. Whereas control and *shTMED3* or *shSOX12* showed only minor differences in the total numbers of spheroids (90 for control, 72 for *shSOX12*, and 78 for *shTMED3* in triplicates with *P* > 0.05 in both cases as compared with controls), the types of spheroids formed varied (Fig 6).

CC14 cells formed two types of spheroids: large and small spheroids (of about half the volume of the large ones), and both were detected at a frequency of ~15% (Fig 6A and B). Cells with compromised SOX12 function showed a reduction by half in the number of large spheres while maintaining the same number of small spheroids formed as compared with controls (Fig 6A and B), arguing that SOX12 is required for the robust formation of large spheroids.

In contrast, *shTMED3* produced a ~10-fold decrease in large spheres and normal small spheres were reduced in number to about one-third (Fig 6A and B). The majority of *shTMED3* spheroids were therefore small but displayed a novel phenotype. These spheroids were loose, with live single cells protruding from the clone and sometimes being clearly separated from it (Fig 6A and B). Both SOX12 and TMED3 thus appear required for normal stem cell clonogenicity, affecting the different types of spheroids formed, but only TMED3 is also required for normal clone compaction/adhesion.

### Knockdown of *TMED3*, but not of *SOX12*, promotes spreading

To test for changes in cell behavior, we first used hanging drop aggregates of cells placed on top of a collagen cushion. Control and *shSOX12* cells remained in the aggregates, whereas *shTMED3* cells spread over the cushion forming a continuous mantle (Fig 7A and B). Isolated cells with elongated morphology leaving the mantle were not generally detected.

To extend this assay, we performed a two-dimensional scratch assay with cell cycle arrested mitomycin C-treated cells, thus allowing the separation of invasion and proliferation. *shTMED3*, but not *shSOX12* cells, showed increase gap closure as compared with control cells (Fig 7C and D). *shTMED3* cells did not generally migrate singly into the gap. Rather, they formed up to 3-fold larger protrusions into the gap region (Fig 7D), indicating the invasive behavior of the cell population.

### Profiling *TMED3* and *SOX12* during tumor progression

Data mining from public platforms revealed that the mRNA levels of *TMED3*, but not of *SOX12*, increased in colon adenocarcinoma

**Figure 6. TMED3 and SOX12 are required for efficient clonogenic behavior but only TMED3 for clone compaction.**

A  Images of representative clonal colonospheres observed under Nomarski (left panels) or GFP fluorescence (right panels) illumination. Arrows indicate cells protruding from the sphere clone after interference with TMED3. Scale bar = 100 μm.

B  Quantification of sphere phenotypes by category presented as the average number of clonal spheroids per 96-well plate, plating 1 cell per well. Overall, the total number of spheroids formed per category in triplicate experiments were (28,32,30 for *SIH* control; 21,22,29 for *shSOX12*; 25,25,28 for *shTMED3*).

versus matched normal colon tissue (Supplementary Fig S6). Similarly, analyses of unmatched samples confirmed significant differences for *TMED3* mRNA but revealed an increase for one of two

**A**

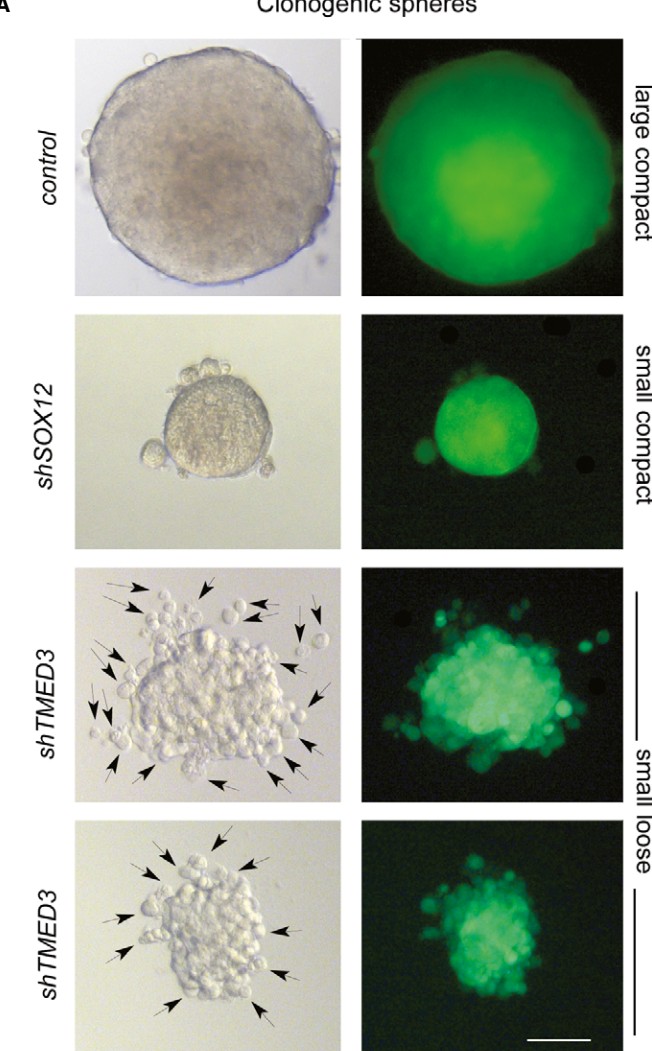

Clonogenic spheres

**B**

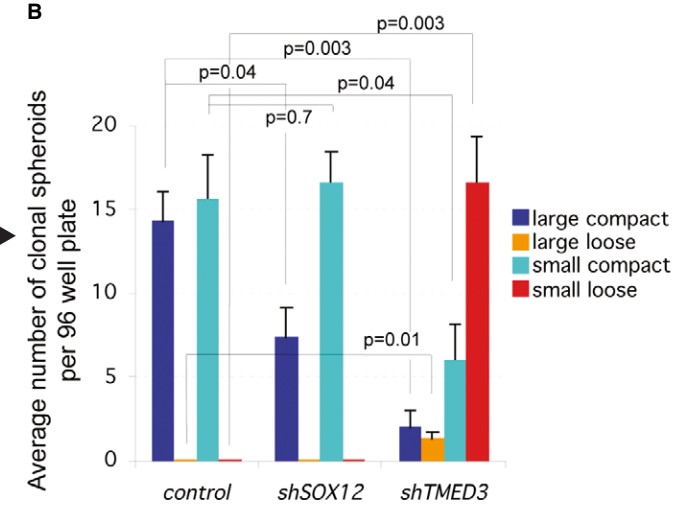

**A**    Hanging drop cell aggregates on collagen

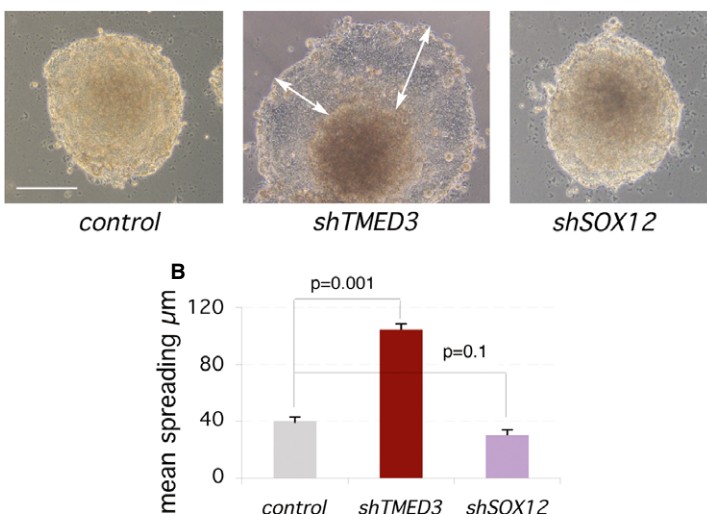

*control*          *shTMED3*          *shSOX12*

**B**

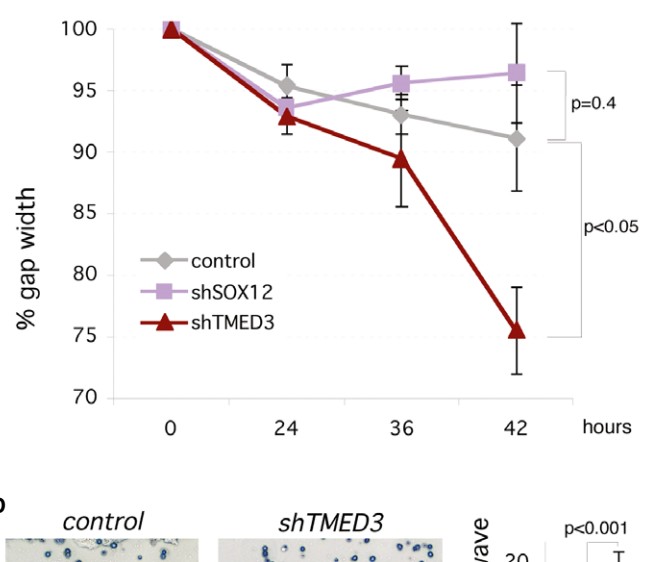

**C** Wound healing assay with mitomycin C-treated cells

**D**

Figure 7.    **Knockdown of TMED3, but not SOX12, induces cell spreading.**

A    Images of aggregates of CC14 cells 5 days after plating on a collagen cushion. The white arrows highlight the cell spreading over the cushion from the aggregate observed only in *shTMED3* cells. Scale bar = 120 μm.

B    Quantification of cell spreading over collagen as shown in (A).

C    Quantification of gap closure over time of experiments shown in (D).

D    Images of the border zone of scratch assays with adherent cells at the end of the assay (42 h). White double-headed arrows and dashed lines indicate the size and limits of protrusions or cell wave invading the gap (left and center panels). The quantification of the number of cells per maximal wave or protrusion is shown on the right panel. c: control; shT: *shTMED3*; shS: *shSOX12*. Scale bar = 300 μm.

    

probes for *SOX12* mRNA (Supplementary Fig S7). Limited data mining did not reveal changes during tumor progression or differences in gender (Supplementary Fig S7).

RT-qPCR assays for *TMED3* and *SOX12* with normal colon and colon cancer patient samples of different TNM stages plus liver metastases obtained from the operating room (Supplementary Fig S8; Varnat *et al*, 2010) confirmed that *TMED3* mRNA was consistently expressed at higher levels in tumors over normal colon, or over normal liver in the case of liver metastases (Supplementary Fig S6). However, *TMED3* mRNA levels were not significantly different in advanced vs. early stages (Supplementary Fig S6). It remains possible that regulation of these genes is post-transcriptional, mimicking effects of shRNAs.

### Interference with *SOX12* or *TMED3* represses endogenous WNT-TCF targets

*SOX12* encodes a SOXC class nuclear transcription factor the function of which has remained largely unaddressed (Sarkar & Hochedlinger, 2013). SOX factors have been described as positive or negative modulators of WNT-TCF signaling, acting on the β-CATENIN/TCF complex (Zorn *et al*, 1999; Sinner *et al*, 2007; Martinez-Morales *et al*, 2010; Kormish *et al*, 2010). In particular, another member of the SOXC family also expressed in colon cancer, SOX4, is a positive cofactor for WNT-TCF signaling (Sinner *et al*, 2007; Kormish *et al*, 2010), raising the possibility that SOX12 could share this activity.

TMED3 is one of 10 members of the transmembrane emp24 domain-containing TMED family of p24 proteins, which are involved in bidirectional cargo trafficking from the rough endoplasmic reticulum through the Golgi and into secretory vesicles (Strating & Martens, 2009). Several *TMED* genes are expressed in colon cancer cells (data not shown), but little is known of the nature of specific cargoes (Strating & Martens, 2009) or the functions of individual p24 proteins, although there is evidence for roles in development (Denzel *et al*, 2000) and the regulation of the innate immune response (Doyle *et al*, 2012). Interestingly, several p24 family members (*Emp24, Eclair, Opossum*) were identified in *Drosophila* as factors required for secretion of Wnt proteins (Wingless, WntD; Port *et al*, 2011; Buechling *et al*, 2011). Although the homolog of *TMED3* in *Drosophila* (*logjam;* Strating & Martens, 2009) may not be involved in Wnt signaling (Carney & Taylor, 2003), we tested for this possibility in human colon cancer cells since p24 proteins are multifunctional (Boltz *et al*, 2007; Strating & Martens, 2009) and TMED5 affects WNT1 secretion in 293T cells (Buechling *et al*, 2011).

Analysis of the expression of a canonical WNT-TCF signaling signature in CC14 colon cancer cells by RT-qPCR after expression of either *shTMED3* or *shSOX12* showed that both induced a drastic reduction in TCF target gene expression levels, largely paralleling the effects of TCF blockade with dnTCF4 (Fig 8A). Commonly repressed genes included the colon and colon cancer stem cell genes *ASCL2* and *LGR5* (Barker *et al*, 2007; Van der Flier *et al*, 2007; Zhu *et al*, 2012) as well as *EPHB2, SOX4* and *P21*, all as compared with control cells after normalization with housekeeping genes (Fig 8A). *AXIN2* was also repressed by *shTMED3*, but not by *shSOX12* although this may reflect different kinetics of repression. Similarly, *shTMED3*, but not *SOX12* enhanced the expression of the *WNT*

inhibitor *DKK1*. The mRNA levels of *β-CATENIN* were not altered and served as an additional control (Fig 8A). These changes parallel those obtained after WNT-TCF pathway blockade by expression of dnTCF4 (Fig 8A), indicating a wide and coherent, albeit partially distinct, downregulation of canonical WNT-TCF responses by knockdown of TMED3 or SOX12. Mimicry of the response to dnTCF4, our benchmark, revealed similar signature changes with *shTMED3* and *shSOX12* in HT29 colon cancer cells (Supplementary Fig S9), albeit with context specific changes and small variations, suggesting a conserved function.

Since downregulation of WNT-TCF targets has been linked to increased HH-GLI activity in colon cancer (Varnat *et al*, 2009), we tested for the expression of key readouts of activity and mediators of this pathway (*GLI1, GLI2*) as well as the expression of the GLI-interacting stemness factor *NANOG* (Zbinden *et al*, 2010) and three other members of a HH-GLI-regulated embryonic stem cell-like signature: *SOX2, KLF4* and *OCT4* (Clement *et al*, 2007; Varnat *et al*, 2010). We observed the coherent upregulation of the expression levels of these genes after knockdown of SOX12 or TMED3, or indeed expression of dnTCF4 (Fig 8A). Knockdown of SOX12 or TMED3 therefore mimics direct repression of WNT-TCF signaling by dnTCF4.

### Enhanced levels of active β-CATENIN rescues the repression of TCF targets by *shSOX12* and *shTMED3*

To further ascertain the specificity of the repression of a TCF target signature by interference with *TMED3* or *SOX12*, we performed rescue analyses testing if active β-CATENIN could revert the repression obtained through the knockdown of these two metastatic suppressors. Co-expression of constitutively active N'Δβ-CATENIN along with *shTMED3* or *shSOX12* from integrated replication-incompetent lentivectors greatly enhanced the expression of a cohort of TCF targets, reverting their repression by *shTMED3* or *shSOX12* alone, both in CC14 and in HT29 colon cancer cells (Supplementary Fig S9). We note that presence of context-dependent differences in the regulation of individual targets. For instance, *SOX4* is repressed by *shSOX12* and *shTMED3* in CC14 cells, but they enhance its expression in HT29 cells, in both cases reproducing dnTCF effects (Fig 8, Supplementary Fig S9). In addition, we find specific target gene differences comparing the regulation by dnTCF versus *shSOX12* or *shTMED3*. For instance, *AXIN2* is repressed by dnTCF in both CC14 and HT29 cells but only by *shSOX12* and *shTMED3* in the latter (Fig 8, Supplementary Fig S9). Notwithstanding cell type-dependent variances, the general rescue of the *shSOX12*- or *shTMED3*-driven repression of TCF targets by activated β-CATENIN, together with their mimicry of dnTCF (see above), argues for a general effect of the knockdown of SOX12 or TMED3 on canonical WNT-TCF signaling in human colon cancer cells.

### SOX12 modulates β-CATENIN/TCF transcriptional activity

As a nuclear transcription factor, we tested for the activity of SOX12 in TCF-dependent dual luciferase reporter assays. Knockdown of SOX12 resulted in a ±50% decrease in endogenous TCF activity (shown as TOP/FOP levels after Renilla normalization; Fig 8B), and increase of SOX12 levels by transfection of its full-length cDNA resulted in a > 2-fold dose-dependent increase in TCF activity when active β-CATENIN was also transfected (Fig 8B). Similar

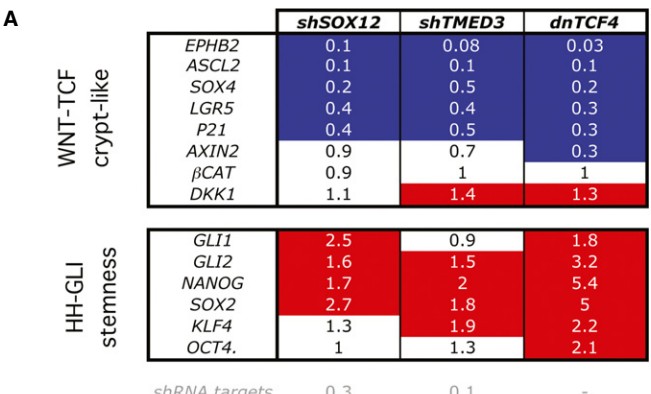

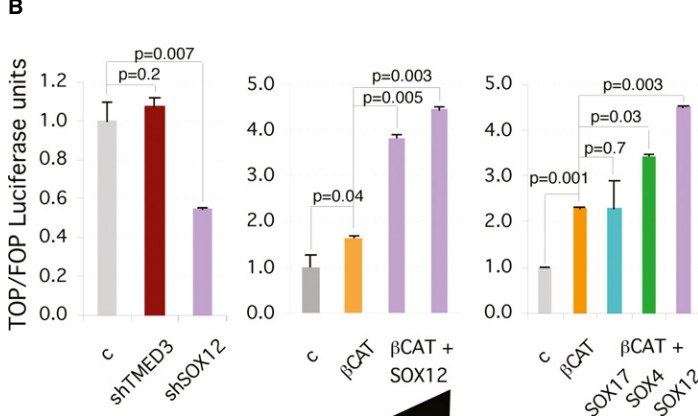

**Figure 8.   TMED3 and SOX12 are required for the expression of endogenous WNT-TCF signaling and SOX12 modulates β-CAT/TCF-driven transcription.**

A   RT-qPCR heat map of the expression levels of WNT-TCF, HH-GLI and stemness gene signatures in CC14 cultured *in vitro* with knockdown of *SOX12* or *TMED3*, or the expression of dominant-negative TCF (dnTCF4). Results are fold expression as compared to the levels in cells infected with control vectors. Red and blue highlighted squares denote changes by 40% or more relative to control values.

B   Representative histograms of luciferase assays for WNT-TCF activity using the TCF-binding site TOP-Luc reporter, and the mutant TCF-binding site FOP-Luc as a control for specificity in CC14 cells. Quantification shows levels as TOP/FOP ratios with the control condition set at 1. The black triangle indicates increasing amounts of *SOX12* plasmid (500 and 1000 ng). βCAT = NΔ-β-CATENIN. 500 ng was used for β-CAT, and 200 ng for SOX17, 4 and 12. Error bars = ±s.e.m.

results were obtained in DLD-1 colon cancer cells (data not shown). Given that these results parallel those with SOX4 (Sinner *et al*, 2007), we compared the activity of equivalent doses of SOX4, SOX17 and SOX12. As expected (Sinner *et al*, 2007), SOX17 had no effect and served as a negative control (Fig 8B). In contrast, both SOX12 and SOX4 (Sinner *et al*, 2007) increased β-CATENIN-dependent TCF reporter activity (Fig 8B).

**TMED3 affects intracellular WNT localization**

To test whether TMED3 KD modifies intracellular WNT localization, we chose to analyze the localization of exogenous WNT1, a *WNT* gene product that is expressed in colon cancer (Holcombe *et al*, 2002). Immunofluorescent detection of transfected, tagged WNT1 revealed a wide distribution of this WNT protein in large vesicles in control CC14 cells (Fig 9A). Labeling was detected both in the soma and in cell extensions, where individual bodies could be easily detected in maximal projections of the confocal z-stack (Fig 9A), or in single confocal sections (Fig 9B–D) using colabeling with Phalloidin to demark the cell bodies (Fig 9B). Such WNT distribution could

correspond to WNT-containing multivesicular bodies (Gross *et al*, 2012; Luga *et al*, 2012).

Localization of endogenous TMED3 using a specific antibody showed that it largely co-localized with transfected WNT1 although domains of single expression were always detected (Fig 9B). Knockdown of *TMED3* resulted in the loss of large WNT-labeled bodies (Fig 9A and B) and the presence of small puncta surrounding the nucleus (Fig 9A), suggestive of ER retention. Co-labeling transfected WNT1 and the trans-Golgi marker TNG46 or the ER marker CALRETICULIN confirmed that WNT1 is largely present in the ER, with a small amount localized to the Golgi (Fig 9C and D).

Previous work on p24 function indicates that blocking ER-to-Golgi trafficking results in the accumulation of components of and proteins secreted through the classical secretory pathway in the ER, including the p24 proteins themselves (Kuiper *et al*, 2001). We thus utilized Brefeldin A to disrupt the Golgi complex (Lippincott-Schwartz *et al*, 1989), blocking normal ER-to-Golgi trafficking, and analyzed the localization of WNT1 in treated vs. untreated cells. Brefeldin A treatment produced a relocalization of WNT1 from large bodies to a diffuse halo of small puncta surrounding the nucleus

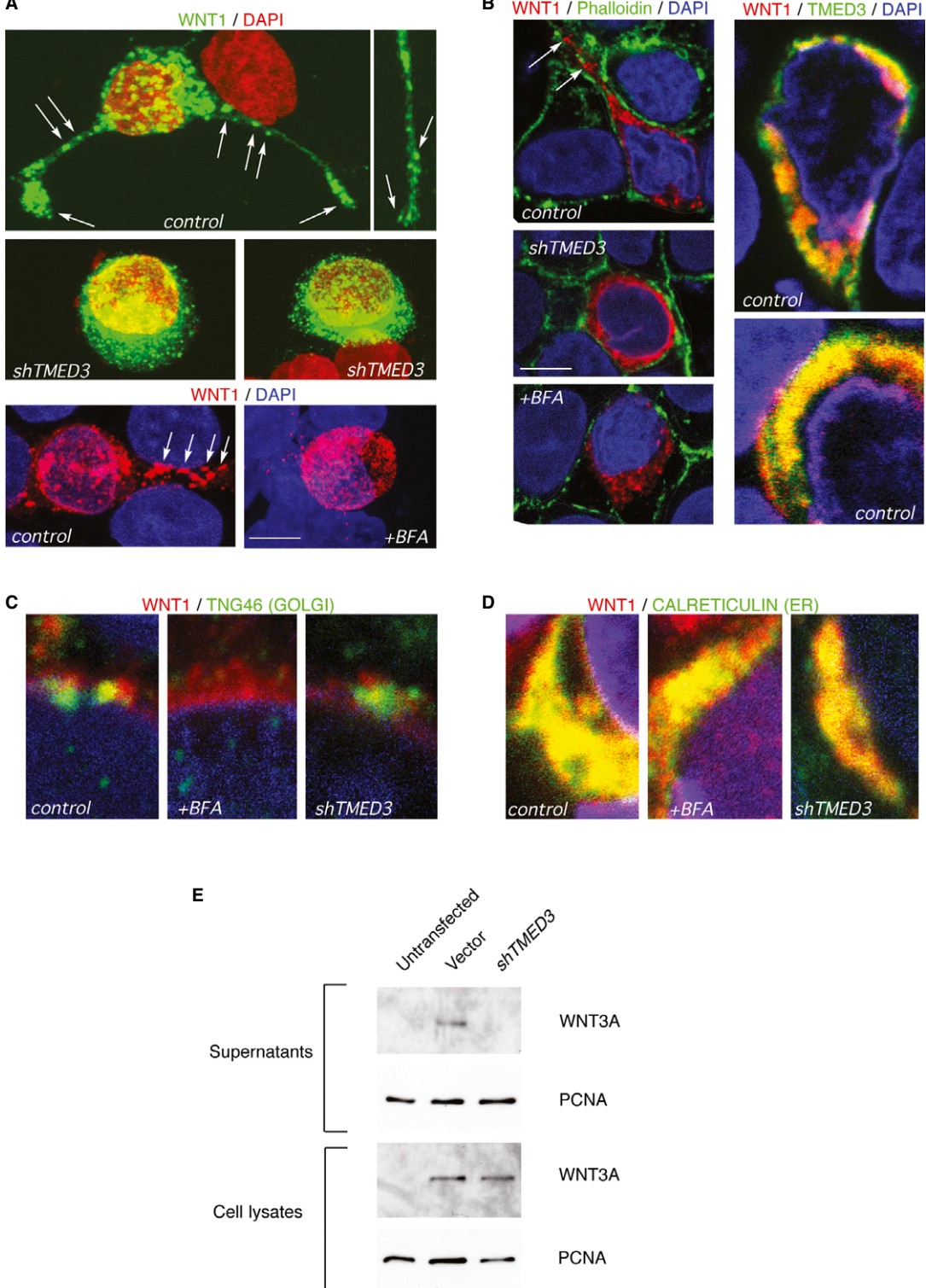

**Figure 9.  TMED3 is required for normal WNT localization.**

A–D  Confocal pictures showing the indicated immunofluorescent labeling in CC14 cells infected with control *shTMED3* lentivectors, or treated with Brefeldin A (BFA).
Images in (A) are maximum projection of confocal Z-stacks. Images in (B-D) are single confocal planes. Yellow color indicates a co-localization of red (WNT1) with green (TMED3, TGN46 or CALRETICULIN) signals. White arrows indicate WNT1-containing vesicular bodies. Cell contours are highlighted with a white line in (B top left). Scale bar = 5 μm (A, B top left 3 panels), 2 μm (B top right 2 panels), 1 μm (C, D).

E  Western blots of V5-tagged human WNT3A protein detected in control and *shTMED3*-expressing cell lysates but only in control supernatants. PCNA was used as control for total lysates as well as for supernatants, as these contain a small amount of intracellular proteins normally released during normal cell death.

(Fig 9A and B). Importantly, this mimicked the effect of interference with TMED3 function. The effects of Brefeldin A in disrupting the Golgi but not the ER were confirmed by the loss of TNG46 labeling (Fig 9C) and the intact distribution of CALRETICULIN (Fig 9D). Together, these results suggest that TMED3 function is required for normal intracellular localization of WNT ligands.

### TMED3 affects WNT secretion

Humans have 19 WNT genes, and deciphering the specificity of TMED3 for different WNT proteins is difficult given the low levels of endogenous expression and the lack of specific high-affinity antibodies for most of them. Nevertheless, to address the ability of TMED3 KD to affect WNT secretion, we have adopted a similar protocol to that utilized with *Drosophila* p24 proteins (Port *et al*, 2011) using exogenous V5-tagged human WNT3A. 293T cells previously infected with *shTMED3* were subsequently transfected with WNT plasmids and total cell lysates as well as supernatants collected and subjected to Western blotting. WNT3A was detected at equivalent levels in cell lysates of control vector and *shTMED3*-expressing cells (Fig 9E). However, it was only detected in the supernatant of control cells and not in those with TMED3 KD (Fig 9E), suggesting the requirement of endogenous TMED3 for WNT secretion.

## Discussion

### A novel *in vivo* genome-wide screen for metastatic suppressors

Elucidating how metastases are normally prevented could yield novel and useful insights both for understanding cell plasticity and for therapeutic purposes. However, defining the cellular functions that inhibit metastases endogenously remains non-trivial, mainly due to the difficulty to track and quantify cells that metastasize *in vivo* and the lack of *in vitro* systems that recapitulate the multiple steps involved in this process. Solely relying on *in vitro* culture is also problematic as this imposes artificial environments such as plastic attachment and high serum conditions. Moreover, many studies have been performed with cloned cell lines, which are only distantly related to patient metastases and do not recapitulate the heterogeneity of primary tumors. These problems have so far hampered accessible genome-wide analyses of relevant endogenous metastatic suppressor functions.

Here, we have designed and implemented a novel assay to bypass the problems raised above and to identify *in vivo* metastatic suppressor functions in an unbiased manner and in a genome-wide scale. In this assay, tagged human cells are directly injected into the venous circulation of host mice to allow efficient lung (or other organ) seeding. Metastatic growth is measured directly or by X-Gal staining of *lacZ*$^+$ transgenic cancer cells and individual shRNAs identified and quantified. Hits are then tested individually in seeding and in full metastatic tests from xenografts to distant organs, notably the lungs. This assay can be adapted to any tumor type and can include loss- or gain-of-function approaches and, importantly, can reach genome-wide representation in each injected animal, thus allowing saturation screens with multiple biological replicates. Our present protocol may also be modified by performing deep

sequencing from tissues with metastases, identifying and quantifying human sequences and thus bypassing time-consuming cloning. Moreover, this assay is also amenable for cell competition *in vivo*, for instance, using the Red/Green assay we previously developed (Varnat *et al*, 2009; Zbinden *et al*, 2010).

### TMED3 and SOX12 encode novel metastatic suppressive functions in human colon cancer cells

To implement our novel screening approach, we have used primary colon cancer cells as proof of principle and have tested the hypothesis, derived from our previous work (Varnat *et al*, 2010), that at least some colon cancer metastatic suppressors should directly or indirectly enhance or support endogenous WNT-TCF signaling.

Our present results define novel functions of the *TMED3* and *SOX12* genes as *in vivo* suppressors of human colon cancer metastases in mice. These two validated candidates were the two top hits based on their representation in independent lung slices with metastases (PCR$^+$ slices) and, importantly, on their presence in 4 of 5 mice (see Fig 1). These results prove that our method can identify the same shRNAs in individual mice carrying a multiple of the genome-wide shRNA library, indicating that genome-wide saturation is indeed possible.

Both TMED3 and SOX12 repress metastatic growth in the lungs, but only TMED3 also suppresses full metastasis from primary tumors, and this is not due to a simple enhancement of tumor growth. Indeed, primary tumor burden can be associated with metastases in different tumors (e.g. Minn *et al*, 2007), but the correlation is not predictive in colon cancer (Miller *et al*, 1985). Instead, SOX12 and TMED3 appear to affect molecular and cellular mechanisms that affect metastatic behavior.

It remains unclear why other WNT-TCF modulators or pathway components (e.g. Sinner *et al*, 2007), or even other metastatic suppressors (e.g. Castets *et al*, 2011), have not been detected in our screen since the repeated identification of *shTMED3* and *shSOX12* in 4 of 5 mice, each with a 5-fold shRNA library coverage, suggests that the screen approached saturation. One possibility may be that the protocol we used to expand cells before injection might have eliminated genes required for *in vitro* growth, including core WNT signaling components, thus allowing the specific selection of essential *in vivo* metastatic repressors. It is also possible that not all WNT-TCF pathway components are represented by efficient shRNAs in the library. Additionally, and more interestingly, the requirements of primary cancer cells may be distinct from those of different cell lines routinely used in previous studies, each with a different genetic background also adapted to *in vitro* culture through thousands of passages. For example, given its reduced potency, it is unlikely that we would have detected TMED3 in a primary screen using HT29 cells.

### SOX12 and TMED3 modulate WNT-TCF in different ways

In addition to unrelated activities of each factor, we find that TMED3 and SOX12 have a common function in the positive regulation of WNT signaling that leads to the sustain expression of TCF targets. Their regulation of a WNT-TCF signature mimics the overall regulation by dnTCF, and their repression is rescued by activated β-CATENIN. Their mode of action, however, would appear to be

radically different. TMED3 encodes a p24 protein belonging to a family previously implicated in vesicular trafficking (Strating & Martens, 2009) and WNT secretion (Port *et al*, 2011; Buechling *et al*, 2011), and SOX12 encodes a nuclear transcription factor of the SOXC family of β-CATENIN/TCF regulators (Sinner *et al*, 2007). Consequently, we suggest that TMED3 acts in WNT protein secretion high in the signaling cascade, whereas SOX12 functions in TCF target gene control at the opposite end.

Our findings contrast with the assumption that both early and metastatic human colon cancers harbor functionally hyperactive WNT-TCF signaling through activation of steps within the signaling cascade, often through loss of APC (e.g. Clevers & Nusse, 2012), and with the finding that WNT signaling is a driver of metastases in other cancers (e.g. Nguyen *et al*, 2009; Yu *et al*, 2012). Our present findings through an unbiased screen support the idea that WNT-TCF signaling represses metastases (Varnat *et al*, 2010): They are consistent with the downregulation of WNT-TCF target gene expression in metastatic vs. non-metastatic human intestinal adenocarcinomas, and with the enhanced metastatic growth of human colon cancer cells with TCF blockade via dnTCF (Varnat *et al*, 2010). We suggest that the common function of SOX12 and TMED3 as positive WNT-TCF modulators *in vivo* underlies, at least partially, their metastatic suppressor functions.

Common APC loss in colon cancer, such as in HT29, might suggest total independence from WNT ligands for pathway activation and thus the inability of blockade of upstream ligand events to affect metastases. Indeed, whereas both the KD of SOX12 and TMED3 are effective in CC14 (with undetermined APC status in this non-clonal primary colon cancer cell population), TMED3 KD has diminished potency in comparison with KD of SOX12 in HT29. However, the metastatic suppressor function of *shTMED3* in HT29 cells, albeit reduced in comparison with CC14, appears consistent with the recent finding that ligand signaling is still required for full WNT-TCF pathway activity and responses in APC-mutant colon cancer cells (Voloshanenko *et al*, 2013).

Colon cancers express multiple canonical and non-canonical *WNT* genes (Dimitriadis *et al*, 2001), and it is not clear what degree of specificity may TMED3 harbor for the different human WNT proteins. This might be important as different WNT ligands can trigger different signaling events in a context-dependent manner (e.g. Gao, 2012). In this sense, one possible explanation for the different phenotypes observed in cell invasion and spreading in *shTMED3* vs. *shSOX12* cells, and the finding that interference with *TMED3*, but not with *SOX12* yields full metastases *in vivo*, might be the blockade of canonical (WNT-TCF) and non-canonical (e.g. planar polarity, calcium) pathways by *shTMED3* but only of canonical transcriptional outputs by *shSOX12*.

Defects in classical secretion by interference with *TMED3* suggest a non-cell autonomous phenotype. However, the results of Red/Green competition assays *in vivo* are inconsistent with the simplest scenario since red control cells should have rescued loss of ligand secretion by neighboring green cells expressing *shTMED3*. The primary effect of knockdown of *TMED3* thus appears to be cell autonomous, suggesting autocrine signaling. Interestingly, red control cells not only did not rescue sibling *shTMED3*-expressing green cells but also suffered a disadvantage *in vivo*. This paradox might be related to the recent finding that autocrine WNT signaling

in breast cancer cells may be aided by stromal exosomes (Luga *et al*, 2012). The stroma could promote autocrine WNT signaling and competitive growth disadvantage in whole WNT1-expressing tumors and in the RFP[+] population in red-green assays, but not in *shTMED3*-expressing GFP[+] cells.

P24 proteins have been suggested to form complexes and cross-regulate their levels (e.g. Jenne *et al*, 2002; Strating & Martens, 2009), and at least in yeast there is functional compensation among family members (Marzioch *et al*, 1999), although in mice TMED2 and TMED10 are non-redundant (Denzel *et al*, 2000; Jerome-Majewska *et al*, 2010). Consistent with non-redundant functions in animals, we have only identified *TMED3* even though we have detected all *TMED* mRNAs except that for *TMED10* in the human colon cancer cells used (data not shown). In *Drosophila*, Éclair, Emp24 and Opossum, which are similar to TMED11, TMED2 and TMED5, respectively, but not other p24 proteins, are involved in Wingless (WNT) secretion (Port *et al*, 2011; Buechling *et al*, 2011; Herr *et al*, 2012). WNT ligand function may thus depend on TMED function in multiple species. However, the exact p24 protein involved in WNT secretion is likely to be species-specific since the expansion of the TMED families apparently occurred independently in different lineages (Strating & Martens, 2009). The functions of p24 proteins also appear to be ligand-type specific since Eclair, Emp24 and Opossum affect Wnt, but not Hedgehog signaling in *Drosophila* (Port *et al*, 2011; Buechling *et al*, 2011; Herr *et al*, 2012). Similarly, different mammalian TMED proteins have been associated with distinct signaling pathways (Doyle *et al*, 2012; Connolly *et al*, 2013; Chen *et al*, 2006). Whereas it remains very likely that TMED3 has additional functions and affects many secreted proteins, given that p24 family proteins load different cargoes into Golgi-derived secretory vesicles and that there are less than a dozen p24 proteins (Strating & Martens, 2009), our data suggest a specific requirement for TMED3 in WNT signaling in human colon cancer.

Unlike TMED3, SOX12 is a multifunctional nuclear transcription factor that promotes β-CATENIN/TCF activity. Given that several *SOX* genes, including *SOX2, SOX4, SOX11* and *SOX12*, are expressed in colon cancer cells (data not shown) and that the cofactor activity we define for SOX12 is similar to that of SOX4 (Sinner *et al*, 2007; Dy *et al*, 2008; Sarkar & Hochedlinger, 2013), it is surprising that the knockdown of SOX12 is not compensated by other SOXC subfamily members as it is the case in mouse embryonic development (Hoser *et al*, 2008; Dy *et al*, 2008; Bhattaram *et al*, 2010). Lack of rescue by SOX4 could be related to its repression by *shSOX12*, but SOX4 was not identified in the screen. Whereas it is possible that TCF-independent aspects of SOX12 function contribute to the phenotypes we observe, our data suggest a novel specific requirement for SOX12 in WNT-TCF target regulation and metastases.

### TMED3 and SOX12 modulate cancer stem cell type

The partially overlapping phenotypes detected after interference with SOX12 or TMED3 highlight key steps required for metastases: Changes in cell adhesion or cell group compaction and changes in stemness. SOX12 and TMED3 antagonize metastatic growth and alter the frequency of large clonogenic spheroids. Large spheroids may reflect the presence of a long-term clonogenic founding stem cell (e.g. Suslov *et al*, 2002). This, together with the repression of

the expression of the stem cell-associated genes *ASCL2* and *LGR4,* which characterize the WNT-dependency of intestinal crypts, adenomas and early local carcinomas (Barker *et al*, 2007; van der Flier *et al*, 2007; Zhu *et al*, 2012), raises the possibility that SOX12 and TMED3 sustain a WNT-dependent, large-spheroid-forming, clonogenic stem cell that is non-metastatic. Changes in stem cell type, as suggested by the varying morphologies obtained in clonogenic assays, could therefore underlie or participate in the acquisition of metastatic potential. In the case of TMED3, this change is more extreme as the spheroids loose compaction and single cells can move out of the clone, perhaps paralleling early steps in cancer stem cell dissemination during metastases.

### Knockdown of TMED3 and SOX12 enhance HH-GLI

In addition to repression of WNT-TCF targets, knockdown of TMED3 or SOX12 in primary CC14 cells increases the expression levels of *GLI1/GLI2* and *NANOG*, which, together, indicate an increased HH-GLI signaling response. This HH-GLI signaling signature is similarly enhanced by direct TCF repression, consistent with our previous data on the enhancement of HH-GLI signaling and metastases by direct repression of WNT-TCF function with dnTCF (Varnat *et al*, 2010).

Moreover, blockade of SOX12, TMED3 or TCF also leads to the general upregulation of the levels of the HH-GLI-regulated genes *SOX2, OCT4* and *KLF4* (Clement *et al*, 2007; Varnat *et al*, 2009, 2010; Zbinden *et al*, 2010). This cohort is able to reprogram adult differentiated cells to embryonic-like states (Takahashi & Yamanaka, 2006), and their upregulation could provoke a change in stemness that prompts metastases (Ruiz i Altaba, 2011). In this sense, the switch in predominant morphogenetic pathway target expression (WNT to HH) we detect after interference with SOX12 or TMED3 supports and expands previous results with dnTCF4 (Varnat *et al*, 2010) and the idea that it promotes and/or tracks a change in tumor cell identity from tissue-specific, crypt-like (for colon cancer) to more embryonic-like metastatic states.

In addition to HH-GLI, YAP also promotes metastases (this study, Lamar *et al*, 2012; Chen *et al*, 2012). This is intriguing in light of reports indicating a tight interaction between the WNT-TCF and HIPPO-TEAD/YAP pathways: YAP1 is regulated by the WNT-TCF pathway (Konsavage *et al*, 2012), the HIPPO-YAP1 pathway regulates WNT signaling (Varelas *et al*, 2010), and YAP1 is essential for β-CATENIN-dependent cancers (Rosenbluh *et al*, 2012). Our results suggest a WNT-TCF-independent role of YAP1 in metastatic colon cancer since we show that YAP1 is essential for metastatic growth whereas endogenous WNT-TCF is not (this work; Varnat *et al*, 2010). One possibility is that in metastases, TEAD/YAP1 signaling interacts with HH-GLI instead of WNT-TCF: HH-GLI signaling drives colon cancer metastases (Varnat *et al*, 2009), and its interaction with TEAD/YAP1 could parallel the interaction described in medulloblastoma (Fernandez-L *et al*, 2009).

### Implications for disease progression and therapeutic intervention

Using a novel *in vivo* assay to identify metastatic suppressor functions, we have uncovered that SOX12 and TMED3 limit the metastatic spread and/or growth of human colon cancer cells in mice.

These findings shed new light on the mechanisms by which primary local tumor cells are restricted from spreading to other organs, but how one may intervene to boost their metastatic suppressive function remains a difficult problem. It also remains unclear whether enhancing their function in metastatic lesions may have a beneficial effect. However, their regulation of WNT signaling suggests important consequences.

Deregulated WNT signaling is a well-known tumor-initiating oncogenic driver in primary colon tumorigenesis (e.g. Kinzler & Vogelstein, 1996). Our present data indicate, paradoxically, that it is also a persistent metastatic suppressor. Therefore, our present and previous findings (Varnat *et al*, 2010) uncover a novel metastatic principle: the repression of the local identity imposed by endogenous tumor-initiating events. This suggests that metastatic cells need to escape the patterning imposed in the local tumor.

Therapeutically, many advanced colon cancers and their metastases are predicted to be refractory to WNT-TCF signaling inhibition (Varnat *et al*, 2010), and this treatment could be counterproductive (this work). Moreover, while adenomas and non-metastatic colon cancers may be negatively affected by WNT-TCF pathway blockade, this could also prompt a few cells to become metastatic, resulting in incurable metastatic disease.

## Materials and Methods

### Cell culture and human tumor samples

CC14 cells are primary TNM4 human colon cancer cells with epithelial morphology (Varnat *et al*, 2009, 2010). All primary colon cancer cells were early passage cells derived from fresh human colon cancer samples obtained directly through Dr P. Gervaz, under approved protocols, from the University Hospital of Geneva (HUG). mRNA was prepared from tumor samples and adjacent normal tissues. Collection, storage, tumor dissociation and primary culture were described previously (Varnat *et al*, 2009, 2010). Human cell lines were cultured as attached cultures in DMEM/F12 supplemented with 10% heat-inactivated FBS in standard conditions with 5% $CO_2$.

### ShRNA library and lentivectors

The shRNA library (GeneNet, human 50K; System Biosciences, Mountain View, CA, USA) was received as a lentivector mix. The lentiviral production was made following the manufacturer's instructions. To control the representation of individual sequences in the lentiviral library, cells were infected with the library and fragments containing the shRNA sequences were amplified by PCR using SIH backbone-specific GNH primers (F: TGCATGTCGC-TATGTGTTCTGGGA, R: CTCCCAGGCTCAGATCTGGTCTAA) and Phusion® DNA polymerase (New England Biolabs). The PCR products were gel-purified, A-tailed with GoTaq® DNA polymerase (Promega) in a single 20′ incubation at 72°C in the presence of dNTPs. A-tailed PCR products were cloned using a TOPO® TA cloning kit (Invitrogen) and transformed into MACH1™ bacteria, allowing the individualization of the PCR products. The plasmid inserts of 50 colonies were sequenced, and the sequences obtained compared to the shRNA list provided by the library manufacturer to

identify the candidate shRNAs. As all 50 of them were different, there is a statistical probability of 95% that the lentiviral library we produced contained at least 46,800 different sequences following the formula $P = [(n - (k - 1)) / n]^{k-1}$ where $n$ is the number of total sequences and $k$ is the number of sequences analyzed.

The shRNAs identified in the *in vivo* screening were in the pSIH1-H1-copGFP (SIH) lentivector (System Biosciences). A second shRNA in the pGIPZ backbone (Open Biosystems) was used to verify specificity and to control for backbone effects. *shYAP1* was cloned in pLVCTH backbone as described (Clement *et al*, 2007). A N'Δβ-CATENIN lentivector in pCWXPG backbone is a kind gift of P. Salmon (University of Geneva).

ShRNA target sequences were 5′–3′ *TMED3:* SIH-GAAGGCATCCGACTGCATTAAGTGTGC, GIPZ- CACCATCTACAGAGAAACGAA; *SOX12:* SIH-TCACACACACAAATCTCAGGAACAAAC, GIPZ-CCAGATTCAAGTCCGTGTGAT; *TEAD1:* SIH-GCCATGGGACATTTACAGCCTTTATAC, GIPZ-GCTCAAACACTTACCAGAGAA; *YAP1:* LVCTH-CGCTGGTCAGAGATACTTCTTAA, *RCD1:* SIH-CCTCAATGCTGAACCGCACTGGAGAA, GIPZ-ACACTGGTTTGGCTTATATAT.

All lentiviruses were prepared and concentrated as described (Stecca *et al*, 2007; Varnat *et al*, 2010).

## BrdU incorporation assays

Human primary cells were infected with LV-*shTMED3*, LV-*shSOX12* or LV-control. 48 h after transduction, GFP-expressing cells were plated in 2% FBS to assess cell proliferation. After a 15-min BrdU pulse, cells were fixed in 4% PFA for 1–2 min, incubated for 15 min in 2N HCl and neutralized in 0.1M Boric acid at pH 8.5 for 10 min. Cells were subsequently blocked in 10% heat-inactivated goat serum and labeled with mouse anti-BrdU antibodies (1:5,000, 2 h at rt; University of Iowa Hybridoma Bank), rhodamine-coupled secondary anti-mouse antibodies (1:500, 1 h at rt; Invitrogen) and counterstained with DAPI to highlight nuclei DNA and allow cell counting (Sigma). Ten fields per condition were counted for quantification and the data expressed as BrdU/DAPI ratios.

## Candidate pro-metastatic shRNA identification

To identify the candidate pro-metastatic shRNAs, lungs from injected mice were dissected and cut into pieces as described in the text. Total DNA was extracted from each piece using a classical lysis followed by a NaAc/EtOH precipitation protocol. The shRNA present in the samples was then identified following the approach described in the shRNA library description paragraph above. At least 20 colonies were sent for plasmid sequencing for each initial lung piece. Sequences obtained were then compared to the shRNA list provided by the library manufacturer to identify the candidate pro-metastatic shRNAs.

## Tail vein injections

Attached cells were trypsinized and resuspended in HBSS, and 300 μl containing $1 \times 10^6$ cells were injected into the lateral tail vein of mice. Cells remaining in the syringe were put back in culture in order to confirm the viability of the injected cells. Mice were analyzed 6 weeks after the injection or at first sign of discomfort.

## Mouse xenografts of human colon cancer primary cells

Human *lacZ*[+] colon cancer primary cells and cell lines were infected with appropriate lentiviral vectors. 72 h after transduction, $5 \times 10^5$ cells were resuspended in HBSS and injected subcutaneously into the flanks of 6- to 8-week-old female nude (NMRI) or NSG (NOD-scid *IL2rg*[null]) mice. Tumor volumes were measured every 2 days after their appearance. Fluorescence of xenografts was monitored in situ in a dark chamber using a green fluorescent excitation laser and a color CCD camera (Lightools, CA, USA). Before tumors reached legal limits, the animals were sacrificed, tumors harvested and lungs were fixed in PFA 4% for 2 h and stained for the β-Galactosidase expression through the X-Gal reaction.

## In vivo Red/Green assays

Red/Green assays *in vivo* were performed as previously described (Varnat *et al*, 2009; Zbinden *et al*, 2010; Lorente-Trigos *et al*, 2010). They involved infecting primary colon cancer cells with either an RFP-expressing control lentivector or GFP-expressing LV-*shTMED3* and LV-*shSOX12* lentivectors. Red (RFP[+]) and green (GFP[+]) populations were then mixed at a constant ratio (approximately 1:1) and injected subcutaneously into the flanks of nude mice. After imaging and dissection, tumors were dissociated and cells FACS-analyzed to quantify the RFP[+] and GFP[+] green populations. Each analysis counted at least 10,000 cells.

## Scratch assays

Human primary colon cancer cells infected with LV-*shTMED3*, LV-*shSOX12* or parental LV-control vectors were plated in 2% FBS medium in 3-cm dishes. When cells reached 80%–90% confluence, they were treated with 10 μg/ml mitomycin C for 2 h to permanently block proliferation, the medium was then changed and several parallel scratches were performed using a blunt spatula 1 mm wide. The result was quantified measuring the gap distance between the edges of the scratch at the time of scratching (time 0 h) and at after each 24 h for 4 days in 3 independent fields per condition.

## Immunofluorescence

Where indicated, Brefeldin A (Sigma, B5936) was added at 5 μg/ml for 2 h. Vehicle DMSO was used as a control. For TGN46 immunofluorescence, cells were fixed with 4% PFA in PBS for 5 min, followed by acetone permeabilization for 5 min at −20°C. Blocking was performed in PBS with 5% HINGS for 45 min. Rabbit anti-TGN46 antibodies (Abcam ab50595) were incubating overnight diluted 1/500 at 4°C in blocking buffer followed by a 45-min incubation with a Dylight488-conjugated anti-rabbit (Jackson ImmunoResearch 111-485-003). After DAPI staining, coverslips were mounted in 50% glycerol with PPD as anti-bleaching agent. For rabbit anti-CALRETICULIN antibodies (Abcam ab2907), the same protocol was followed except that the permeabilization was performed with methanol at −20°C for 10 min. For rabbit anti-TMED3 antibodies (NOVUS Biologicals, NBP2-13439, 1/100) and mouse monoclonal anti-Flag-tag (Sigma F3165, 1/2,000) immunofluorescence, the same protocol was followed except that permeabilization was performed

with Triton X-100 (0.1%, 10 min). For double immunostaining, mouse anti-FLAG antibodies were added together with other rabbit primary antibodies and revealed with a rhodamine-conjugated anti-mouse secondary antibody (Molecular Probes R-6393, 1/500). Alexa488-conjugated Phalloidin (Invitrogen A12379, 1/300, a kind gift from C. Chaponnier) was incubated together with anti-FLAG primary antibodies and revealed as described above.

**Luciferase assays**

For luciferase assays, $1 \times 10^5$ CC14 cells, naive or infected with the appropriate shRNAs, were plated in 24-well plates and then transfected with Lipofectamine LTX with a plasmid containing TOP/FOP reporters (500 ng), pGL4.74 Renilla luciferase (100 ng, Promega), pRFP as transfection control (100 ng), CMV-β-CATENIN or CMV-SOX4/SOX12/SOX17 as required. A pCMV-control was used to complement or replace the other expression plasmids in order to keep the amount of DNA and of the CMV promoter constant.

**Clonogenic assays**

For *in vitro* clonogenic assays, cells infected with different shRNAs were counted and plated at 1 cell/well in 96-well plates with colonosphere media (DMEM/F12 supplemented with $1\times$ B27 and 10 ng/ml EGF) in triplicates. The number of total and GFP$^+$ colonies per plate was counted after 14 days using an inverted optical Zeiss microscope equipped with epifluorescence. For tumor-derived clonogenic assays, tumors obtained from mouse xenografts were dissociated and resuspended in 1XHBSS to make single-cell suspension. GFP$^+$ human cells were FACS-sorted and automatically deposited at clonogenic 1 cell/well ratios in 96-well plates in colonosphere medium, with 5 plates prepared per each condition. The total number of colonies and that of GFP$^+$ colonies per plate were counted after 14 days as above.

**Cell spreading from aggregates on collagen**

Cells were trypsinized and allowed to aggregate for 2 days in 20 µl hanging drops containing 1,000 cells each in standard culture medium. Aggregates were then transferred onto a cushion of collagen (1 mg/ml; Sigma) in 24-well plates with fresh medium and imaged after 5 days. Medium was changed once after 2 days of transfer. The phenotype was slightly variable depending on collagen batches.

**PCR**

RNA extraction, cDNA synthesis and quantitative reverse transcription (rt) PCRs were performed as described (Stecca *et al*, 2007; Lorente-Trigos *et al*, 2010; Varnat *et al*, 2010). All values were normalized against the mean of those of three housekeeping genes: *TBP*, *HMBS* and *ACTB*. qPCR primer sequences are shown in Supplementary Fig S10.

**Data mining**

Data mining was performed on the following publicly available microarray datasets:

**The paper explained**

**Problem**

Metastatic cancers remain a large unmet medical need. To date, there are no efficient anti-metastatic therapies and most therapies do not differentiate between the primary and secondary metastatic tumors. In the case of colon cancer, over one-third of patient with local adenocarcinomas in the bowel will develop metastases, most commonly to the liver, with an average life expectancy of less than a year. Understanding the mechanisms that normally prevent metastases could yield valuable clues for new therapeutic strategies. However, finding endogenous metastatic suppressors has been hampered by a number of problems, including the use of cell lines and *in vitro* conditions.

**Results**

We have developed a novel *in vivo* strategy to identify metastatic suppressors in human cells engrafted into mice using an shRNA library that allows saturations screens. As proof of principle, we have performed a screen with primary human colon cancer cells. Moreover, we tested the hypothesis derived from our previous work that at least some tumor suppressors should promote WNT signaling in human colon cancer. We report the identification of two genes, SOX12 and TMED3, the suppression of which causes an increase in metastatic growth in the lungs or full metastases from a primary tumor. TMED3 functions in protein secretion through the ER-Golgi and SOX12 is a nuclear transcription factor. Importantly, both support endogenous WNT signaling, apparently acting at opposite ends of the signaling cascade.

**Impact**

Our screening method should be widely applicable to single patient tumors and to multiple tumor types, opening up the systematic identification of metastatic suppressors. Moreover, we identify two previously unknown metastatic suppressor functions for primary human colon cancer cells that share a common signaling function: Both support the expression of WNT-TCF targets. Therefore, in addition to highlighting TMED3 and SOX12 as novel metastatic suppressors, the data provide additional evidence that the suppression of endogenous WNT signaling is pro-metastatic. As WNT-TCF activity is a common oncogenic tumor-initiating event in colon carcinogenesis in humans, these findings highlight a novel metastatic principle: suppression of the tumor-initiating event. Whereas WNT-TCF antagonists may reduce early primary tumor bulk, they are also predicted to enhance metastases. Thus, the present results support the differential targeting of primary colon cancers and their metastases.

— GSE10950, composed of 24 colon normal and tumor (Jiang *et al*, 2008). Probe for *TMED3* was ILMN_1719316, probe for *SOX12* was ILMN_1736974. Description and files are available at http://www.ncbi.nlm.nih.gov/geo/query/acc.cgi?acc = GSE10950

— GSE23878 composed of 24 normal colon samples and 35 unpaired colon tumor (Uddin *et al*, 2011). Probe for *TMED3* was 208837_at, probes for *SOX12* were 204432_at and 228358_at. Description and files are available at http://www.ncbi.nlm.nih.gov/geo/query/acc.cgi?acc = GSE23878

— GSE6988, composed of paired tissues of 25 normal colorectal mucosa, 27 primary colorectal tumors, 13 normal liver and 27 liver metastasis, as well as 20 primary colorectal tumors without liver metastasis (Ki *et al*, 2007). Probes for *TMED3* were ID_ref 10974 and 11770, probes for *SOX12* were ID_ref 9506 and 13496. Description and files are available at http://www.ncbi.nlm.nih.gov/geo/query/acc.cgi?acc = GSE6988

— GSE17538 composed of 238 tumor samples for which the clinical follow up of the patient is available (Smith *et al*, 2010; Freeman *et al*, 2012). Probe for *TMED3* was 208837_at, probe for *SOX12* was 204432_at. Description and files are available at http://www.ncbi.nlm.nih.gov/geo/query/acc.cgi?acc = GSE17538

**Detection of WNT protein in the supernatant**

$1.5 \times 10^6$ of 293T cells infected with GIPZ-shTMED3 or control lentiviruses were seeded in 6-cm dishes in DMEM with 10% FBS. The following day the infected cells were transfected using calcium phosphate with 0.8 μg of V5-tagged WNT3A expression plasmid (Addgene 35927). Medium was replaced 6 h after by 4 ml of DMEM with 0.5% FBS. Supernatants were collected 48 h later, spun twice on a table top centrifuge to remove any floating cells or debris and concentrated ~20-fold (Pierce Concentrators 9K MWCO, 89884A, Thermo Scientific). Cells were also collected directly from the dish and lysed in RIPA buffer. 25 μg of total protein from the concentrated supernatants or 2 μg from the cell lysates were loaded per well onto 12% SDS–PAGE. Western blotting was performed using anti-V5 (ab27671, Abcam, 1/500 overnight) and anti-PCNA (SC-7907, Santa Cruz, 1/1,000 1 h) antibodies.

**Supplementary information** for this article is available online: http://embomolmed.embopress.org

## Acknowledgements

We are grateful Monika Kuciak, Grigori Singovski, Carolina Bernal, Irene Siegl-Cachedenier and other members of the Ruiz i Altaba lab for discussion or comments on the manuscript, Dr P. Gervaz (HUG) for colon cancer samples, Drs J. Kiss and P. Salmon for WNT1- and N'Δβ-CATENIN-lentivectors and Dr C. Chaponnier for conjugated Phalloidin. AD was a recipient of a fellowship from the French Foundation pour la Recherche Médicale. AM was a European ITN FP7 Network HEALING Fellow. SM was a recipient of a grant from the Swiss Confederation. Support for this work was provided by the Swiss National Science Foundation, the European Union FP7 ITN network HEALING, La Ligue Suisse Contre Le Cancer, the James McDonnell Foundation (USA) and the Département d'Instruction Publique of the Republic and Canton of Geneva to ARA.

## Author contributions

ARA, AD, AM, SM and MM designed the experiments. AD, AM, MM, SM, CS and AC performed experiments. ARA, AD, AM, MM, SM and CS analyzed the data. ARA wrote the paper with AD and AM.

## Conflict of interest

The authors declare that they have no conflict of interest.

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
