## [Review Process File · EMBO Molecular Medicine]

A novel genome-wide in vivo screen for metastatic suppressors in human colon cancer identifies the positive WNT-TCF pathway modulators TMED3 and SOX12

Arnaud Duquet, Alice Melotti, Sonakshi Mishra, Monica Malerba, Chandan Seth, Arwen Conod and Ariel Ruiz i Altaba

Corresponding author: Ariel Ruiz i Altaba, University of Geneva

Review timeline:

Submission date:	06 August 2013
Editorial Decision:	13 September 2013
Author rebuttal and resubmission:	23 December 2013
Editorial Decision:	28 January 2014
Revision received:	12 March 2014
Editorial Decision:	07 April 2014
Revision received:	11 April 2014
Accepted:	15 April 2014

Transaction Report:

Editor: Roberto Buccione

1st Editorial Decision

13 September 2013

Thank you for the submission of your manuscript to EMBO Molecular Medicine. We have now heard back from the three referees whom we asked to evaluate your manuscript.

As you will see, all three Reviewers, while recognising the merits of the study, point to significant issues that, I am afraid, altogether preclude publication of the manuscript in EMBO Molecular Medicine. I will not discuss each point in detail as they are clearly stated.

Reviewer 1 recognises the technical prowess and interest of the study but raises fundamental concerns on the overall biological message. S/he notes the lack of focus on a particular gene and its role on metastasis and does not feel that the model proposed to reconcile some of the observations is supported by the data. For instance, Reviewer 1 mentions a crucial issue, i.e. that to conclude that TMED3 and SOX12 are required for stem cell function requires substantially more focused experimentation. S/he also notes that TMED3 and SOX12 are not shown to modulate hedgehog signalling and that the claim that TMED3 affects WNT secretion is not supported fully by the data.

Reviewer 2 is more positive but notes that the study was performed with a single cell type and that,

in any case, crucial information on that cell type is missing.

Reviewer 3 also is of the opinion that the fact that the study was performed with a single cell type is a major weakness. S/he also criticises the *in vivo* metastasis models used to draw conclusions and notes that although primary colon cancer and metastasis data were supplied, these do not support the hypothesis. I should mention that this Reviewer, similarly to Reviewer 1, also has issues with the stem cell aspects of the manuscript.

Reviewers 1 and 3 also list other specific items of concern.

Given these critical issues, I hope you will understand that we have no choice but to return the manuscript to you at this stage.

I am sorry to disappoint you on this occasion. I hope, however, that the Reviewers' comments will be helpful for your continued work in this area.

***** Reviewer's comments *****

Referee #1 (Remarks):

A genome-wide *in vivo* screen identifies TMED3 and SOX12 as suppressors of colon cancer metastases and positive regulators of WNT signaling

In this study by Duquet, et al., two suppressors of colon cancer metastases, TMED3 and SOX12, are identified, and their relationship to the Wnt signaling pathway is elucidated. The group used a lentiviral shRNA library to perform an *in vivo* screen of metastasis using an interesting, novel primary colon cancer cell system. Metastatic tumors were collected and analyzed for shRNAs, from which TMED3 and SOX12 were among the top candidates. Individual knockdown studies were used to confirm that loss of TMED3 and/or SOX12 promote metastasis. The authors used mouse xenograft assays to determine that knockdown of both genes conferred a survival advantage for cancer cells, but only TMED3 induced more metastases compared to nontransduced cells. Knockdown of TMED3 but not SOX12 was also shown to increase cancer cell invasiveness. The authors suggest that TMED3 is involved in the cellular export of Wnt ligands and that SOX12 is involved in Wnt signaling and target gene regulation. The authors conclude that strong (autocrine) Wnt signaling suppresses metastatic behavior. A notable strength of this study is the entirely novel screen developed to identify metastasis regulators. The development of this screen seems like a powerful discovery tool which is applicable to multiple cancer types. In this regard, the study has broad appeal. Additional techniques and cell model systems make this study interesting. However an equally notable concern is that the study is not focused on any one particular gene and its role in metastasis. The attempts to combine two (three) differently acting genes into a cohesive, weakly supported model is not fully supported by the data presented.

Major concerns:

1. The overall finding of this manuscript is based on a shRNA screen that identifies genes whose loss affects metastasis. Three genes are presented, TMED3, SOX12 and TEAD1/YAP1. Part of the issue here is the brief treatment of TEAD1, a knockdown of which causes an opposite phenotype which is a profound loss of metastasis. Knockdown of TMED3 and SOX12 cause a strong increase in metastasis. There is no data presented to show that the opposing phenotypes are related, and the YAP1/TEAD1 analysis is only cursory and a distraction to the main topic of the study. Either the YAP1/TEAD1 data is relegated to supplemental data, or it should be taken out without much loss to the overall manuscript. (a minor note here: the profound loss of metastasis could be due to loss of cell viability - no experiments are shown that test this fundamental possibility). Otherwise, the authors need to better justify why inclusion of TEAD/YAP data is important for the study.
2. The authors show that loss of TMED3 causes loss of "sphere compaction" a visible morphological phenotype of cancer spheroids in which cells appear loosely attached in a non-adherent culture setting. Loss of SOX12 does not produce the same phenotype. In fact, SOX12-negative cells create small, very compact spheroids. This is one of several data presented showing that loss of SOX12 does not always phenocopy loss of TMED3. While this may not necessarily be a surprise - these

genes could function in myriad processes, not all of them related - it does highlight how the Abstract might be overemphasizing a common link between these two genes to WNT signaling.

3. TMED3 and SOX12 are proposed to enhance WNT target gene expression, yet the overall effect on b-catenin and Wnt signaling is not examined - only the listing of a partial set of known WNT target genes.

4. Knockdown of TMED3 leads to loss of cell compaction, and this has an intriguing possible link to the stimulation of metastasis. The loss of cell compaction in spheroids, and the loss of cell extensions in the staining shown in Figure 6 and 9, suggest that there is a loss of cell adhesion. Loss of E-cadherin, would certainly affect cell adhesion, Wnt signaling, and cell survival in clonogenic assays and metastasis (EMT). The authors should explore this possibility since there are published reports of links between SOX proteins (SOX9) and regulation of VE-cadherin expression. Internalization of adhesion complexes is also connected to endosomes and the Golgi and as such might have a link to TMED protein function.

5. The abstract states the data shows that full Wnt signaling inhibits metastasis, a model that links to this group's earlier reports showing the Hedgehog pathway emerges to drive metastasis. Yet, manipulation of TMED3 and SOX12 are not shown here to modulate HH signaling (although the authors do show changes in hedgehog signaling components). If TMED3 and/or SOX12 truly change metastatic behavior, then changes in HH signaling should be evident.

6. The authors state (page 8, top paragraph) that both TMED3 and SOX12 are required for "stem cell function". The concern here is that this conclusion derives mostly from changes in the morphology of cancer spheres (Figure 5). Cancer stem cell function is best studied in terms of survival in single cell/clonogenic assays coupled with an assay that assesses the multipotent status of the cells. As it stands here, the authors have not proven that TMED3 and SOX12 are important for cancer stem cell states.

7. If TMED3 truly affects WNT secretion and not cell shape and extensions, then the authors need to show that WNT ligands are not secreted. The WNT1 staining is intriguing, but could strictly derive from a change in cell morphology as this ligand is overexpressed. Furthermore, the staining is not identical to the Brefeldin A control. Perhaps a better control is the Porcupine inhibitor IWP2, which unlike BrefeldinA, is a specific inhibitor of Wnt processing and secretion. IWP2 could serve as a positive control and a direct comparison to TMED3 knockdown.

Minor concerns:

1. No error bars in wound healing assay (Fig 6C)
2. Some of the text is confusing, particularly in introduction paragraphs

Referee #2 (Remarks):

This study used genome-wide shRNA screen in primary human colon cancer cells and injection of shRNA library-expressing and control cells into nude mice to search for suppressors of colon cancer metastasis. A number of other assays were subsequently performed to evaluate the metastasis suppressor properties of the genes identified.

Two novel metastasis suppressors were identified, TMED3 and SOX12, which were found to function at opposite ends of the WNT-TCF signaling cascade, in secretion of WNT proteins (TMED3) and activation of nuclear transcription (SOX12). Knock-down of TMED3 or SOX12 increased metastatic growth and decreased colon cancer stem cell clonogenicity. Together with an earlier paper from this group (Varnat et al. 2010), the findings highlight the principle that repression of the tumor-initiating event can lead to metastatic progression and may explain why the expression levels of WNT-TCF target genes are often downregulated in advanced colon cancer.

Overall, this study contains important new information and has obvious clinical implications in the development and use of treatment regimens based on blockage of WNT signaling at different stages of colorectal cancer.

Minor comments:

1. A major part of all data are derived from experiments on a single primary colon cancer cell line (CC14), which was chosen because of the morphological and growth properties of the cells. It would be useful to know the basic molecular characteristics of this cell line as well, such as the WNT/beta-catenin activation status, modal chromosome number, and the type of genetic instability (CIN/MIN).
2. It is hypothesized on page 14 that TMED3 and SOX12 might differ in their preferences to function in canonical vs. non-canonical pathways. Are there any concrete data (by the authors or by other groups) to support this hypothesis?

Referee #3 (Remarks):

This interesting manuscript by Duquet et al seeks to identify suppressors of colon cancer metastases using a novel in vivo screen. In general the experiments shown appear well performed and controlled. Although it is difficult to accept their conclusions, given the work of many other laboratories and the focused nature of their own experimental approach, the conclusions reached are consistent with their work shown here and elsewhere. In my opinion, two experimental weaknesses decrease the potential impact of this work. Firstly, their conclusions are really limited to what occurs in one cell line- albeit a primary colon cancer cell line. The second general weakness is their dependence on tail-vein injections and metastases secondary to primary xenografted tumors, as models of tumor metastases. Both of these weaknesses could be mitigated through the analyses of primary colon cancer samples and metastases. And although such data was provided, it did not support their hypothesis. Some more specific points are outlined below:

1. Their conclusions are based on the premise that the identified components are dedicated, primary components of the canonical Wnt signaling pathway. Although they do provide some data to support this premise, it is far from conclusive. Knocking down more generally accepted components of the Wnt pathway would go along way to support their premise. Of some concern, such established components of the Wnt signaling pathway did not show up in their in vivo screen, which they claim to be a saturating screen.
2. Some additional data, and textual clarification, regarding the function of TMED3 would also be helpful. Does knockdown of TMED3 limit the secretion of endogenous Wnts in CC14 cells? Can the biological outcomes resulting from TMED3 knockdown be rescued by the addition of exogenous Wnts? What role do endogenous Wnts play, in an autocrine fashion, in a CRC cell line in which downstream tumor suppressors are removed?
3. The growth curves for the 100% shTMED3 or shSOX12 tumors in nude mice should be shown.
4. If knock down of TMED3 affects CSC self-renewal, why does it not affect the growth of the primary tumor?
5. In the competition experiment between the red and green CC14 cells in vivo, what happens to the red cells? If TMED3 knock down does not result in increased proliferation but does decrease CSC self-renewal, how does it outcompete the control red CC14 cells in vivo?
6. What is the functional difference between increased numbers and size of the metastases observed?
7. Many of the legends lack the detail to understand the figures shown.
8. References for some of the techniques they use were sometimes missing
9. Descriptions of work on TEAD1 and YAP were confusing.
10. On page 8, you twice add a +/- sign as a prefix of a colony description?

December 23rd, 2013

Reply to Referee's and Editor's comments
EMM-2013-03359

Dear Dr. Buccione,

Thank you for sending the comments of the referees and your own. Following our conversation on the telephone, we are glad to be able to address the various issues raised. I trust you will agree that the changes, additions and discussion on various issues should made our paper acceptable for publication in EMBO Molecular Medicine.

Importantly, we have refocused the paper to emphasize the novel screen and provide additional data on the effects of SOX12 and TMED3 KD on metastases in vivo with another colon cancer cell type (new Fig. 3), the regulation of *AXIN2* and other WNT-TCF targets in another cell type (new Fig.S9), and the requirement of TMED3 for WNT secretion (new Fig. 9E).

Editor and Reviewers

As per your letter there were five major issues, which we address below. We then provide a point per point answer for the comments of each referee.

Point 1- Lack of focus on one gene

Reviewer 1 recognises the technical prowess and interest of the study but raises fundamental concerns on the overall biological message. S/he notes the lack of focus on a particular gene and its role on metastasis and does not feel that the model proposed to reconcile some of the observations is supported by the data.

From Referee 1:

A notable strength of this study is the entirely novel screen developed to identify metastasis regulators. The development of this screen seems like a powerful discovery tool, which is applicable to multiple cancer types. In this regard, the study has broad appeal.

In the original version we had mostly focused on the results of the screen and, as such, we reported the two top hits. We realize that the novelty of the screen, nicely recognized by the editor and the referees, merited more explicit presentation. We have thus refocused the paper to present the novel screen and its proof of principle through its implementation with one primary colon cancer. It is thus normal to present the results of such a screen, much as a genetic screen, with all the validated hits and the process to reach these. We thus opt to maintain the presentation of both genes, TMED3 and SOX12, since these were found in 4/5 mice each carrying a multiple of the genome-wide shRNA library, and these were the only validated hits in our near-

saturation screen. Indeed, it is critical to present all the data to validate our approach, including the TEAD/YAP data.

In addition to the screen, we now clearly state that through this novel technology we tested a critical hypothesis derived from our previous work (Varnat et al., 2010), namely, that colon cancer metastatic suppressors in primary tumor cells in vivo should affect WNT-TCF signaling. Our unbiased results support this idea. This is indeed surprising for a number of reasons, including the fact that we have not isolated other possible candidates, other members of the multigene families to which TMED3 and SOX12 belong, or even other positive components of the WNT pathway. However, the reasons why this is so can only be hinted at in the discussion, as we already do. We do not pretend to resolve all issues raised from our present study. To the contrary, our work should engender novel and additional studies.

Point 2- Stem cell data

For instance, Reviewer 1 mentions a crucial issue, i.e. that to conclude that TMED3 and SOX12 are required for stem cell function requires substantially more focused experimentation. I should mention that this Reviewer (3), similarly to Reviewer 1, also has issues with the stem cell aspects of the manuscript.

We thank you and the referees for highlighting the issues with stem cells. We note that whereas is not the major issue of the paper, the referees concerned with this point appear to have misread the old Fig. 5 (now Fig. 6). While Panel A shows the different morphologies, our conclusions are not solely based on these findings. We have quantified the number of spheroids in true clonogenic assays in 96-well plates, as clearly shown in panel B. Indeed, panel B represents in quantitative form both the number and kind of spheroids. It is thus a direct measure of stem cell self-renewal in this assay, which is accepted in all major scientific journals. Moreover, we explicitly provide the p values for each relevant comparison. Our conclusions and discussion on cancer stem cells are clearly related to the experiments performed and are fully supported by the data provided.

Point 3- HH signaling and WNT secretion

S/he also notes that TMED3 and SOX12 are not shown to modulate hedgehog signalling and that the claim that TMED3 affects WNT secretion is not supported fully by the data.

As stated in the text, we track paper activity by target gene signatures as changes in the expression levels of single genes can be misleading. We show that the expression of GLI1, PTCH1 and GLI2 are altered. This allows us to discuss that HH signaling is affected (e.e. Varnat et al., 2009). Moreover, NANOG is a GLI target (Zbinden et al., 2010) and it is also altered. Such signatures are used in many contexts, including cancer classifications and transcriptomic data interpretation. We maintain that based on the signature provided in Fig. 8, the response to HH signaling is altered. Similarly, changes

in a WNT-TCF signature allow us to conclude that WNT-TCF is altered (Fig. 8). Importantly, these changes mimic the direct regulation of target genes by genetic means through the expression of dnTCF. We thus respectfully disagree with this referee and maintain that signatures, as shown in Fig. 8, allow us to conclude that both WNT and HH signaling are altered. This is a major point of the paper. We recognize, however, that this may not have been well presented and we have modified the text accordingly.

In contrast, we agree with the note that our data did not directly address whether WNT secretion is affected by TMED3. Indeed, we provided data to only suggest this point (Fig. 9) and referred to previous work in *Drosophila* that indicate such activity (Port et al., 2011; Buechling et al., 2011). In human cells we are confronted with the daunting task to test 19 WNT proteins with less than optimal antibodies at mostly very low expression levels. Defining which endogenous WNT is affected will require a lot more experimentation and analyses and these fall outside of the scope and focus of the present paper. Indeed, the main findings are: 1- the in vivo screen data and 2- the modulation of WNT-TCF signatures by shTMED3 and shSOX12 that mimic dnTCF4.

Nevertheless, we have worked very hard to produce data that would address the points raised by this referee. We now provide new data with a second colon cancer cell type, HT29, showing that KD of SOX12 or TMED3 largely mimics the changes produced by dnTCF (new Fig. S9). Notably, *AXIN2*, a bona fide direct TCF target, is repressed to the same extent in *shSOX12*, *shTMED3* and *dnTCF4* conditions.

In addition, we now provide data (new Figure 9E) that shows that endogenous TMED3 function is required for secretion of V5-tagged WNT3A.

All the new data support our initial conclusions.

Point 4- Single cell type.

Reviewer 2 is more positive but notes that the study was performed with a single cell type and that, in any case, crucial information on that cell type is missing. Reviewer 3 also is of the opinion that the fact that the study was performed with a single cell type is a major weakness.

We acknowledge the criticism of the referees and have endeavored to provide data in other colon cancer cells to support our findings. Whereas the screen took several years, as we developed it and sorted through the candidates, we note that we used a single primary human colon cancer as proof of principle. Nevertheless, we agree that the key question is if the function of TMED3 and SOX12 in regulating WNT-TCF targets is conserved.

We provide new in vivo data to support a more general role in suppressing metastases in the new Fig. 3 using HT29 colon cancer cells.

Point 5- In vivo assays.

S/he also criticises the in vivo metastasis models used to draw conclusions and notes that although primary colon cancer and metastasis data were supplied, these do not support the hypothesis.

We are confounded by these comments. We have used the best in vivo metastatic assays available in two immunocompromised hosts. There are no proven orthotopic grafts for colon cancer. Rectal wall injections lead to bleeding and direct entry of grafted cells into the circulation. The tumor-to-distant organ metastases we use are highly controlled and reproducible and recapitulate full metastases. We are unsure of what assay could this referee be thinking about. We have consulted our medical colleagues and no one came up with better grafting approaches. Many papers provide data in a single mouse model and a single metastasis protocol. We provide data with two models and two protocols.

Moreover, we indeed provide primary colon and metastases data but the data we can mine with enough samples are extremely limited. Data mining shows that there is little relevant data sorted by TNM stage and most arrays are done with whole tumors without knowing what the contribution of non-tumor tissues may be. Nevertheless, we have used cohorts previously used in other publications and show the results in old Fig.7 (now Fig. S6). Our own PCR data with patient samples corroborates the findings, and this is also shown in Fig. S6. We note that the data does not show a reduction in the expression levels of *TMED3* or *SOX12* in early versus late tumors or, in a limited fashion, in metastases as compared with advanced local tumors, but this cannot possibly invalidate our data or approach. As explained in the new version, it may well be that there are post-transcriptional changes that affect these genes, or a myriad of other possibilities. Interestingly, post-transcriptional changes would, in fact, mimic the effects of shRNAs in our experimental setting. Notwithstanding these issues, it seems difficult to accept that if a simple possibility, namely that *TMED3* or *SOX12* mRNAs are not downregulated, this should necessarily invalidate our findings. We respectfully disagree with the remarks in Point 5 above and submit that 1- we use the best metastatic assays in vivo available, and 2- the transcriptional data provided in Figure S6 cannot in any way invalidate our data and findings.

Referee #1:

We thank this referee for her/his detailed comments and the care in reviewing our work. The general issues have already been incorporated into the comments above.

Major concerns:

1. The overall finding of this manuscript is based on a shRNA screen that identifies genes whose loss affects metastasis. Three genes are presented, TMED3, SOX12 and TEAD1/YAP1. Part of the issue here is the brief treatment of TEAD1, a knockdown of which causes an opposite phenotype which is a profound loss of metastasis. Knockdown of TMED23 and SOX12 cause a strong increase in metastasis. There is no data presented to show that the opposing phenotypes are related, and the YAP1/TEAD1 analysis is only cursory and a distraction to the main topic of the study. Either the YAP1/TEAD1 data is relegated to supplemental data, or it should be taken out without much loss to the overall manuscript. (a minor note here: the profound loss of metastasis could be due to loss of cell viability - no experiments are shown that test this fundamental possibility). Otherwise, the authors need to better justify why inclusion of TEAD/YAP data is important for the study.

As mentioned above, the new version first presents the screen, its novelty and its implementation, before moving to the study of the two validated hits and the testing of the hypothesis that metastatic suppressors in human colon cancer in vivo should affect WNT signaling. Therefore, we think that it is critical to include all controls and steps in the screen. In this sense, we feel that it is important to discuss the TEAD/YAP approach and results. We note that there is already supplementary material but we opt to keep the figures of TEAD/YAP in the main body of the paper since these also serve as control for the experiments with TMED3 and SOX12. For instance, seeing the opposite effect for YEAP in Red/Green in vivo assays is very reassuring given the similar phenotype in these tests for both TMED3 and SOX12. Finally, for this point, we note that all injected cells are viable and thus this simple possibility was indeed tested. It remains possible, indeed, that the long term effect of YAP knockdown is loss of viability, but this referee will agree that this is not the place to expand on the mechanism of action of YAP.

2. The authors show that loss of TMED3 causes loss of "sphere compaction" a visible morphological phenotype of cancer spheroids in which cells appear loosely attached in a non-adherent culture setting. Loss of SOX12 does not produce the same phenotype. In fact, SOX12-negative cells create small, very compact spheroids. This is one of several data presented showing that loss of SOX12 does not always phenocopy loss of TMED3. While this may not necessarily be a surprise - these genes could function in myriad processes, not all of them related - it does highlight how the Abstract might be overemphasizing a common link between these two genes to WNT signaling.

It is clear that knockdown of TMED3 and of SOX12 do not produce identical phenotypes. No one would have thought this to be likely. It is also clear that p24 and SOXC family proteins are multifunctional. What is exciting is that both

were picked up in a screen testing for metastatic suppressors and that both downregulate a WNT-TCF signature, mimicking dnTCF. Moreover, they both reduce clonogenicity but only TMED affects compaction and full metastases. They do have partially overlapping functions. While we are very explicit about the multifunctionality of these classes of proteins in the discussion, one important possibility that we discuss is that the partially diverse phenotypes are due to the fact that they appear to act at the opposing poles of the WNT-TCF pathway. Thus, simple possibilities that we discuss in the paper are that their differences could be due to branching pathways downstream of ligand action, e.g. canonical versus non canonical. Resolving these issues remains outside of the scope of this paper but the fact that TMED3 and SOX12 show partial differences in their knockdown phenotypes is not only normal but expected given their natures.

3. TMED3 and SOX12 are proposed to enhance WNT target gene expression, yet the overall effect on b-catenin and Wnt signaling is not examined - only the listing of a partial set of known WNT target genes.

This referee is technically correct but we do not agree the premise that an extended TCF target signature is not predictive. This has been used in hundreds of publications in all kinds of journals. Moreover, β CATENIN is a multifunctional protein (also involved in adhesion and it affects other pathways) and it would be difficult to interpret protein changes. For example, changes in SOX12 affect TCF reporters in the presence of β CATENIN (Fig. 8B), thus dissociating β CATENIN levels and TCF output.

4. Knockdown of TMED3 leads to loss of cell compaction, and this has an intriguing possible link to the stimulation of metastasis. The loss of cell compaction in spheroids, and the loss of cell extensions in the staining shown in Figure 6 and 9, suggest that there is a loss of cell adhesion. Loss of E-cadherin, would certainly affect cell adhesion, Wnt signaling, and cell survival in clonogenic assays and metastasis (EMT). The authors should explore this possibility since there are published reports of links between SOX proteins (SOX9) and regulation of VE-cadherin expression. Internalization of adhesion complexes is also connected to endosomes and the Golgi and as such might have a link to TMED protein function.

We thank this referee for this very helpful and insightful comment. We are certainly following this kind of lead but this paper is not the place to expand on function of TMED3 in these interesting areas.

5. The abstract states the data shows that full Wnt signaling inhibits metastasis, a model that links to this group's earlier reports showing the Hedgehog pathway emerges to drive metastasis. Yet, manipulation of TMED3 and SOX12 are not shown here to modulate HH signaling (although the authors do show changes in hedgehog signaling components). If TMED3

and/or SOX12 truly change metastatic behavior, then changes in HH signaling should be evident.

We respond to this comment above but to restate our answer, we argue that the signature changes (of GLI1/2 and NANOG) are clearly indicative of changes in HH-GLI signaling. This kind of evidence is used in many fields, including cancer and pathway analysis in transcriptomics. Moreover, in this case, we have the direct targets of the pathway and mediators, as well as a functionally relevant GLI target. Together, we feel confident this signature can be correctly interpreted to mean that HH-GLI signaling is modified. Moreover, as mentioned by this referee, the changes we observe in the HH-GLI signature after expression of dnTCF match those seen with shSOX12/shTMED3 shown in Fig. 8, and those previously reported in Varnat et al., 2010 for dnTCF, where we build on our work on the HH pathway (Varnat et al., 2009). However, to make sure our points are clear, we have modified the text to exactly reflect the results presented.

6. The authors state (page 8, top paragraph) that both TMED3 and SOX12 are required for "stem cell function". The concern here is that this conclusion derives mostly from changes in the morphology of cancer spheres (Figure 5). Cancer stem cell function is best studied in terms of survival in single cell/clonogenic assays coupled with an assay that assesses the multipotent status of the cells. As it stands here, the authors have not proven that TMED3 and SOX12 are important for cancer stem cell states.

We fully agree with this referee. As discussed above, s/he seems to have misread the histogram in old Figure 5B (now Fig. 6B), which exactly reports the requested data.

7. If TMED3 truly affects WNT secretion and not cell shape and extensions, then the authors need to show that WNT ligands are not secreted. The WNT1 staining is intriguing, but could strictly derive from a change in cell morphology as this ligand is overexpressed. Furthermore, the staining is not identical to the Brefeldin A control. Perhaps a better control is the Porcupine inhibitor IWP2, which unlike BrefeldinA, is a specific inhibitor of Wnt processing and secretion. IWP2 could serve as a positive control and a direct comparison to TMED3 knockdown.

We thank this referee for this careful point. However, it is entirely possible that changes in cell morphology are a consequence of abnormal WNT signaling. We do not see why they need to be necessarily separate. The staining cannot be identical to Brefeldin A treated cells since in this case all classical secretion is compromised. Instead, the phenotypes are similar (but not identical). We also thank this referee for the helpful comment to use IWP2. However, in our hands neither this nor C59 work very well or reproducibly in control cells and have thus opted to avoid using these, unfortunately-not-so-clean, pharmacological agents.

As mentioned above, we now provide data showing that endogenous TMED3 is required for the secretion of V5-tagged WNT3A (new Fig. 9E).

Minor concerns:

1. No error bars in wound healing assay (Fig 6C)

We thank this referee for noticing this important detail. They were left out by mistake. They have now been added in the new Fig. 7C.

2. Some of the text is confusing, particularly in introduction paragraphs

As mentioned above we have rewritten the paper and refocused. We trust the new version will clarify any possible previous confusion.

Referee #2:

Overall, this study contains important new information and has obvious clinical implications in the development and use of treatment regimens based on blockage of WNT signaling at different stages of colorectal cancer.

We thank this reviewer for her/his positive comments.

Minor comments:

1. A major part of all data are derived from experiments on a single primary colon cancer cell line (CC14), which was chosen because of the morphological and growth properties of the cells. It would be useful to know the basic molecular characteristics of this cell line as well, such as the WNT/beta-catenin activation status, modal chromosome number, and the type of genetic instability (CIN/MIN).

We agree with this referee that CC14 was chosen for its clear epithelial morphology and its stable phenotype in vivo and in vitro. However, we would like to underline that this is *not* a cell line. It has never been cloned and thus retains heterogeneity. It is thus difficult to see how one would go about determining chromosome number reliably, etc. However, we have applied the latest criteria for subdivision of primary colon cancer types by the group of Hanahan (Sadanandam et al., 2013). We find that CC14 is of the Transit amplifying inflammatory type. This information is now included in the paper and shown in Fig S1.

2. It is hypothesized on page 14 that TMED3 and SOX12 might differ in their preferences to function in canonical vs. non-canonical pathways. Are there any concrete data (by the authors or by other groups) to support this hypothesis?

This referee is correct in stating that we raise the possibility of differential effect of TMED3 and SOX12 knockdown based on the idea that WNT secretion high in the pathway could affect multiple branches, notably canonical and non-canonical pathways. However, this is only a discussion item and sorting this will required heavy additional experimentation outside of the scope of this paper. We note that there are no described reliable, universal targets for non-canonical signaling.

Referee #3 (Remarks):

This interesting manuscript by Duquet et al seeks to identify suppressors of colon cancer metastases using a novel in vivo screen. In general the experiments shown appear well performed and controlled.

We thank this referee for highlighting the novelty of and confidence in our screen and approach. We now emphasize this point in the paper.

Although it is difficult to accept their conclusions, given the work of many other laboratories and the focused nature of their own experimental approach, the conclusions reached are consistent with their work shown here and elsewhere.

We respectfully disagree with the first sentence of this referee, which leads us to think that s/he is not unbiased towards our work. While s/he acknowledges the consistency of our work, s/he appears somewhat predisposed against it, staining the rest of the review.

In my opinion, two experimental weaknesses decrease the potential impact of this work. Firstly, their conclusions are really limited to what occurs in one cell line- albeit a primary colon cancer cell line.

We agree with this referee that providing additional data should make our study more general and have discussed this comment above. We now provide additional data on the function of TMED3 and SOX12 as WNT-TCF regulators in another colon cancer cell type.

The second general weakness is their dependence on tail-vein injections and metastases secondary to primary xenografted tumors, as models of tumor metastases.

We respectfully disagree with this referee. What are better experimental models in vivo? We address this issue also in the first set of comments above but would like to restate that there are no better *experimental* approaches available over those we have used here in two strains of immunocompromised mice. To be honest, we are not sure what this referee requests.

Both of these weaknesses could be mitigated through the analyses of primary colon cancer samples and metastases. And although such data was provided, it did not support their hypothesis.

We disagree with his/her assessment. The first weakness this referee notes is resolved with additional data. The second is not a weakness. Notwithstandingly, we provide additional data mining analyses and direct PCR

measurements of gene expression levels in a collection of patient tumors obtained from the OR. The results of measuring gene expression levels in patient samples cannot possibly invalidate the functional results. Moreover, the 'hypothesis' is not whether TMED3 and SOX12 transcripts disappear, but that functional metastatic suppressors should positively sustain or boost WNT-TCF signaling. This we prove. How this comes about in patients is to be resolved. We note that shRNAs could mimic post-transcriptional effects that would not be obvious in transcript analyses. The fact that the simplest explanation is not supported by transcript data does not mean the functional data is not valid or significant.

Some more specific points are outlined below:

1. Their conclusions are based on the premise that the identified components are dedicated, primary components of the canonical Wnt signaling pathway. Although they do provide some data to support this premise, it is far from conclusive. Knocking down more generally accepted components of the Wnt pathway would go along way to support their premise. Of some concern, such established components of the Wnt signaling pathway did not show up in their in vivo screen, which they claim to be a saturating screen.

In addition to the screen a key result in our present study is that knocking down TMED3 or SOX12 drastically reduces WNT-TCF target gene expression to the same extent as pan-dominant negative dnTCF. Certainly, another possible strategy for our work would have been to knockdown each component of the WNT-TCF pathway but this was not our approach, as clearly stated in the revised version: We have developed and implemented an unbiased and novel screen in vivo for metastatic suppressors. We claim that this screen approached saturation since we identified the same shRNAs in 4/5 independent tests, which parallels picking up multiple alleles in a genetic screen. Not picking up other components of the WNT pathway is not necessarily a concern, rather it is of great interest! Should it be a concern that mutations in *BRAF*, *KRAS*, *PTEN*, *AKT1* and other genes are not picked up each time in each cancer type or cancer sample? We do not think so. To the contrary, it is interesting as it relates to the very nature of cancer development, pleiotropism and context-dependency. The fact that we have not detected other WNT pathway regulators in our screen is likely related to these very same issues and it is interesting and certainly not a matter of concern. One cannot second-guess a screen when multiple alleles are detected.

2. Some additional data, and textual clarification, regarding the function of TMED3 would also be helpful. Does knockdown of TMED3 limit the secretion of endogenous Wnts in CC14 cells? Can the biological outcomes resulting from TMED3 knockdown be rescued by the addition of exogenous Wnts? What role do endogenous Wnts play, in an autocrine fashion, in a CRC cell line in which downstream tumor suppressors are removed?

This referee points to very interesting possibilities. As discussed in the first part, resolving which of the 19 WNTs is affected by TMED3 knockdown is not a straightforward matter, given both the lack of specific antibodies for all of them and the low levels of expression of many of them. Similarly, it is not clear which WNTs could rescue the effect and if 'exogenous' WNT as supernatant, for instance, could rescue vesicle-bound WNT or other forms. These are long and complex experiments out of the scope of this paper but we certainly agree that and interesting and merit exploration. Regarding the last point we note that a very interesting and recent paper by the Boutros group indicates that ligand-driven WNT signaling is required even in the case of loss of APC (Voloshanenko et al., 2013). This indicates that ligand signaling in colon cancer with loss of APC, for instance, is fully relevant, further supporting our findings. This reference is now added to the paper. Importantly, we now show that TMED3 is required for the secretion of V5-tagged WNT3A.

3. The growth curves for the 100% shTMED3 or shSOX12 tumors in nude mice should be shown.

For these experiments we have added histograms of final tumor weights in Fig. S3B.

4. If knock down of TMED3 affects CSC self-renewal, why does it not affect the growth of the primary tumor?

This is an excellent point to explore further. We suggest that it may affect cell behavior rather than proliferation per se as the bulk of the tumor is not made by CSCs, TMED3 knockdown appears to affect the self-renewal of CSCs (with only a partial decrease in total sphere numbers as shown the new Fig. 6B) but not the proliferation of resulting progenitors, as suggested from BrdU analyses (Fig. S3). Clearly, further experiments are possible but these are out of the scope of the present paper. However, at this point a simple explanation is that whereas TMED KD affects CSCs in vitro, where they are WNT-TCF-dependent, it may not affect tumor growth in vivo since CC14 tumors in mice (unlike CC14 cells in vitro) are not affected by TCF blockade (Varnat et al., 2010).

5. In the competition experiment between the red and green CC14 cells in vivo, what happens to the red cells? If TMED3 knock down does not result in increased proliferation but does decrease CSC self-renewal, how does it outcompete the control red CC14 cells in vivo?

This is an excellent point that we discuss openly in the text. We are not sure what mechanisms are at work here. Further experimentation is required but this is, again, outside of the scope of this paper.

6. What is the functional difference between increased numbers and size of the metastases observed?

We measure both given that they could relate to distinct phenomena. Colony number is a measure of seeding while colony growth is a measure of local behavior of cells. In doing so we highlight the possible differences in the function of TMED3 and SOX12. Indeed, knockdown of the first but not the second leads to seeding of the lungs from a distant tumor, whereas both affect the growth of directly seeded colonies.

7. Many of the legends lack the detail to understand the figures shown.

We have revised the entire paper and trust the legends are now appropriate.

8. References for some of the techniques they use were sometimes missing.

We have tried to be exhaustive in our referencing. We have revised the paper but specific comments would have been helpful here.

9. Descriptions of work on TEAD1 and YAP were confusing.

We have rewritten the paper and trust the descriptions of all experiments are clear.

10. On page 8, you twice add a \pm sign as a prefix of a colony description?

We have changed the +/- sign for the sign ~ to indicate the 15% and 10-fold are 'rounded' numbers. The readers can look at the figures for precise data in the graphs.

We thank all referees for their constructive comments and suggestions for improvement.

Thank you for the submission of your manuscript to EMBO Molecular Medicine. We have now heard back from the three Reviewers whom we asked to evaluate your manuscript.

You will see that while all three Reviewers are supportive of your work, Reviewers 1 and 3 express a number of concerns that prevent us from considering publication at this time. I will not dwell into much detail, as the evaluations are detailed and self-explanatory.

Reviewer 1 is primarily concerned about the over-reaching conclusions drawn from the available data. Although substantial improvement compared with the original version is acknowledged, s/he does note that some problems do persist in this respect. Reviewer 1 also challenges your statement concerning the specificity of Porcupine inhibitors; I should add that I especially share his/her comment on the inadequacy of Brefeldin A in this context. Finally, Reviewer 1 notes that the conclusions on the role of TMED3 and SOX12 in stem cell function require appropriate quantification of the number of spheroids that develop in each condition. My impression is that this Reviewer is globally quite positive but does raise sensible points; I would therefore ask you to place a special effort to address the above concerns scrupulously and with additional experimentation where necessary.

Reviewer 3, similarly to Reviewer 1, also urges you to be more cautious in the interpretation of your experimental findings. Clearly this issue must be solved. S/he also notes that an important control rescue approach is required to establish specificity of the TMED3 and SOX12 RNAi. This Reviewer also lists a number of other points for your action.

While publication of the paper cannot be considered at this stage, we would be pleased to consider a suitably revised submission, with the understanding that the Reviewers' concerns must be fully addressed with additional experimental data where appropriate and that acceptance of the manuscript will entail a second round of review.

Please note that it is EMBO Molecular Medicine policy to allow a single round of revision only and that, therefore, acceptance or rejection of the manuscript will depend on the completeness of your responses included in the next, final version of the manuscript.

As you know, EMBO Molecular Medicine has a "scooping protection" policy, whereby similar findings that are published by others during review or revision are not a criterion for rejection. However, I do ask you to get in touch with us after three months if you have not completed your revision, to update us on the status. Please also contact us as soon as possible if similar work is published elsewhere.

I look forward to seeing a revised form of your manuscript as soon as possible.

***** Reviewer's comments *****

Referee #1 (Comments on Novelty/Model System):

A genome-wide in vivo screen identifies TMED3 and SOX12 as suppressors of colon cancer metastases and positive regulators of WNT signaling

In this revised manuscript Duquet and colleagues further clarify the new in vivo screen for proteins that affect metastasis of cancer cells. As stated in the previous round of review, the invention of the screen is the main strength of the manuscript. The focus of this first iteration of the screen is colon cancer and two of the identified hits from the screen are validated and studied for the mechanism by which they might influence metastasis. While there were over-reaching conclusions about the mechanisms by which TMED3 and SOX12 influence metastasis and strong statements that they are Wnt signaling components, this current version provides a more tempered assessment. Overall this is an improved version of the study. Specific comments are provided below.

1. One of the main improvements of this manuscript is an important re-write of the Abstract to eliminate the original language that stated two of the identified hits from the screen, TMED3 and SOX12 are "required for endogenous WNT-TCF target expression". Such a definitive statement requires definitive proof, and while the authors provide interesting and intriguing data about TMED3 and SOX12, the data do not fully support that statement. In the current Abstract, the authors have re-worked the language to offer a more realistic summary: TMED3 and SOX12 "promote" WNT-TCF signaling.

2. Additional data that pertain to the influence on WNT-TCF signaling are provided (Fig.8 and S9). TMED3 and SOX12 affect Wnt signaling and target gene expression, but in variable and moderate levels. For example, the data show a mild change, at best, on canonical WNT activity (as measured by TOPflash), and interesting but variable effects on endogenous target gene expression. These influences indicate that TMED3 and SOX12 have complex effects on oncogenic Wnt signaling. These data also support the notion that TMED3 and SOX12 are not central protein players in Wnt signal transduction. One case in point, comparison of the dnTCF4 signature shows overlapping effects on gene expression for some known Wnt targets, but paradoxical effects on P21 (gene name CDKN1A). P21 is a G1 cell cycle inhibitor that increases dramatically when dnTCF4 is expressed (van de Wetering et al., 2002). Other groups that express dnTCFs also see p21 expression increase sharply - a finding not replicated here. This difference hints at a level of complexity above and beyond a simple model that TMED3 and SOX12 regulate direct WNT targets - and merits a measured, accurate conclusion statement in the Abstract and treatment in the Discussion.

3. TMED3 action: In response to reviewer requests the authors now provide a stronger piece of data to show that knockdown of TMED3 prevents secretion of an epitope-tagged Wnt ligand. This data strengthens their conclusions about TMED3 action.

4. That the authors provide an improved direct test with an epitope-tagged Wnt and have satisfied the concern from this reviewer is one thing, however this reviewer wishes to clarify a point made in the author's rebuttal. In response to the request by this reviewer that a specific inhibitor of Wnt ligand secretion be used instead of Brefeldin A to examine WNT ligand expression, the reviewers declined this suggestion stating that Porcupine inhibitors do not work in their hands, and that they are "not-so-clean pharmacological agents". It is important to clarify this point. IWP2 and C59 are quite specific small molecule inhibitors of Porcupine, an ER-resident enzyme shown by multiple groups to function directly in Wnt signaling. Both inhibitors (and also the closely related LGK974 molecule) work at low nanomolar concentrations. There is also the next-generation IWP, IWP-L6 molecule and these operate at sub-nanomolar levels (Wang X, Moon J, Dodge ME, Pan X, Zhang L, Hanson J, et al. The Development of Highly Potent Inhibitors for Porcupine. J Med Chem. 2013). Brefeldin A blocks all protein transport from ER-to-Golgi and in so doing triggers ER stress and the

Unfolded Protein Response (ultimately apoptosis). It is thus NOT valid to rely on a comparison of Brefeldin A and Wnt ligand localization and conclude that TMED3 promotes their secretion.

5. Clonogenic Spheroid Assay for Stemness: Clonogenic survival in an anchorage-independent growth assay is certainly one way that aspects of stemness can be evaluated. But this reviewer does not see how the data in Fig. 6B is being misread. The percentage of each type of spheroid that develops under each condition is reported (both in the figure, panel B and in the legend). The text verbally restates the data shown in Fig. 6B. However, there is no place in the manuscript where the absolute number of spheroids that develop in each condition are given. Instead, the authors state that 15% of wildtype CC14 cells survive to develop spheroids. But then they then state on page 9 that:

"...Cells with compromised SOX12 function showed a reduction by 50% in the number of large spheres but not of smaller ones (Fig. 6A,B). In contrast, shTMED3 induced a drastic ~10-fold decrease in large spheres with the majority (mostly small spheres) had a loose phenotype, with live single cells protruding from the clone and sometimes being separated from it (Fig. 6A,B). Both SOX12 and TMED3 thus appear required for efficient stem cell function but only the latter is also required for normal cell group compaction/adhesion...."

Stating the relative proportion of large versus small spheroids says nothing about stemness (alho this metric is interesting for what it might say about metastasis). This point is belabored here because the authors make a very strong conclusion about the role of TMED3 and SOX12 in both the Abstract and Discussion. What is needed is data showing the actual number of spheroids that develop in each condition.

Referee #2 (Remarks):

In this entirely re-written manuscript, CC14 primary human colon cancer cells of the transit amplifying inflammatory type and HT29 colon cancer cells are used to demonstrate the role of the positive WNT pathway regulators TMED3 and SOX12 in suppressing metastases. The use of two types of colon cancer cells giving concordant results (Fig. 3, Fig. S9) and the revised focus significantly improve the manuscript.

Referee #3 (Comments on Novelty/Model System):

The authors use an innovative screening strategy for regulators of metastasis involving transduction of shRNA libraies followed by in vivo screening for lung metastasis of colon tumor cells. They identify two gene whose knockdown consistently increases metastasis of colon carcinoma cells. The experimental design is well described and experiments are neatly documented. The two candidates could have important functions in the metastatic progress and are therefore of medical interest. Moreover, a negative correlation of metastasis formation and activation of the Wnt pathway and a positive correlation to hedgehog signalling is indicated, in line with previous reports by the authors.

Referee #3 (Remarks):

Major critical points:

1 The enrichment of shRNAs in the primary screening approach towards genes that are not expressed in the injected tumor cells or that appear to act in opposite pathways (shTEAD) is a bit worrying. The authors ascribe these to "off-target effects" which would mean that their system is somehow sensitive to such effects. Although two different shRNAs to TMED3 and SOX12 support specificity the true control would be a rescue approach using transduction of shRNA by insensitive cDNA constructs.

2 Authors should be more cautious in the interpretation of their results. The fact that TMED3 and Sox12 can activate the Wnt pathway does not mean that their effects on metastasis are due to alterations of wnt signalling, although it would fit well with previous data from the group. Unless the authors show that activation of Wnt signalling, e.g. by active b-catenin would rescue effects of shTMED and shSOX12 the connection to Wnt signalling remains rather correlative. Their components could well have other functions in secretion of other factors (TMED3) or in gene expression of wnt unrelated genes (SOX12). This should be stated in the discussion and some of the passages extensively dealing with the Wnt connection might be shortened. It should also be noted that there was no effect of shTMED3 on TOP/FOP reporters which is difficult to reconcile with its effects on Wnt target gene expression.

Minor points:

Is there an explanation why shTMED and shSox12 cells outcompete the control cells in the red/green assay but show no difference to controls in tumor growth when grafted separately?

The use of the term "primary colon cancer" to describe the CC14 cells is misleading as one would assume that tumor tissues have been grafted which was not the case. Instead one could replace this term by "primary colon tumor cells".

What is the meaning of +/- 50% or +/-60-90% used on pages 5 and 6?

At certain points authors should be more precise:

p. 6 line 7 from bottom delete "than"?

p. 8 second paragraph: should probably read "The ability of the knockdown of ..." Similarly, in the abstract: "knockdown of TMED3 ... induces full metastasis..."

p14 third line from bottom: "However, it was only detected in the supernatant of control cells..."

p. 22 third line from bottom: It is useless to show formula if n, k are not described.

Editors comments:

Reviewer 1 is primarily concerned about the over-reaching conclusions drawn from the available data. Although substantial improvement compared with the original version is acknowledged, s/he does note that some problems do persist in this respect.

Please see the point per point response below.

Reviewer 1 also challenges your statement concerning the specificity of Porcupine inhibitors; I should add that I especially share his/her comment on the inadequacy of Brefeldin A in this context.

Brefeldin A is used to show mislocalization of Wnt1 following Golgi disruption. As such it is used correctly in our work. We do not claim that Brefeldin A can be used to test for WNT secretion per se. We have modified the text to make sure this comes across clearly.

Regarding the specificity of Porcupine inhibitors, please see our reply below. We bought the improved IWP6 compound, as suggested by the reviewer, performed three types of experiments and provide you with the data. We note that the paper reporting this compound is a good chemistry paper with limited biology (partial inhibition of zebrafish tail regrowth and very limited biochemistry at high drug levels). There is no direct evidence of effects on primary human colon cancer cells. We regret not being able to be more positive on these compounds but this does not directly affect our submission.

Finally, Reviewer 1 notes that the conclusions on the role of TMED3 and SOX12 in stem cell function require appropriate quantification of the number of spheroids that develop in each condition.

As described below we have taken the necessary steps to resolve the problem with quantification and discussion.

My impression is that this Reviewer is globally quite positive but does raise sensible points; I would therefore ask you to place a special effort to address the above concerns scrupulously and with additional experimentation where necessary.

We thank you for your encouraging words. We have spent a lot of time to address this referee's concerns. Please see our responses below.

Reviewer 3, similarly to Reviewer 1, also urges you to be more cautious in the interpretation of your experimental findings. Clearly this issue must be solved.

We have added extra caution in our paper. We have now reworked the abstract, introduction, results and specially the discussion section. We note that the new rescue data reinforces our original conclusions.

S/he also notes that an important control rescue approach is required to establish specificity of the TMED3 and SOX12 RNAi.

The rescue has been done with active β CATENIN as requested, and the data included in Fig. S9. Please see below.

This Reviewer also lists a number of other points for your action.

Thank you. Please see below.

Referee #1

In this revised manuscript Duquet and colleagues further clarify the new in vivo screen for proteins that affect metastasis of cancer cells. As stated in the previous round of review, the invention of the screen is the main strength of the manuscript. The focus of this first iteration of the screen is colon cancer and two of the identified hits from the screen are validated and studied for the mechanism by which they might influence metastasis. While there were over-reaching conclusions about the mechanisms by which TMED3 and SOX12 influence metastasis and strong statements that they are Wnt signaling components, this current version provides a more tempered assessment. Overall this is an improved version of the study.

We thank this referee for his/her positive comments.

Specific comments are provided below.

1. One of the main improvements of this manuscript is an important re-write of the Abstract to eliminate the original language that stated two of the identified hits from the screen, TMED3 and SOX12 are "required for endogenous WNT-TCF target expression". Such a definitive statement requires definitive proof, and while the authors provide interesting and intriguing data about TMED3 and SOX12, the data do not fully support that statement. In the current Abstract, the authors have re-worked the language to offer a more realistic summary: TMED3 and SOX12 "promote" WNT-TCF signaling.

We thank this referee for his/her comments. As a response to Referee 3 we now also provide the rescue of the WNT-TCF signature with active β CATENIN, the result of which reinforces our original conclusions.

2. Additional data that pertain to the influence on WNT-TCF signaling are provided (Fig.8 and S9). TMED3 and SOX12 affect Wnt signaling and target gene expression, but in variable and moderate levels. For example, the data show a mild change, at best, on canonical WNT activity (as measured by TOPflash), and interesting but variable effects on endogenous target gene expression. These influences indicate that TMED3 and SOX12 have complex effects on oncogenic Wnt signaling. These data also support the notion that TMED3 and SOX12 are not central protein players in Wnt signal transduction. One case in point, comparison of the dnTCF4 signature shows overlapping effects on gene expression for some known Wnt targets, but paradoxical effects on P21 (gene name CDKN1A). P21 is a G1 cell cycle inhibitor that increases dramatically when dnTCF4 is expressed (van de Wetering et al., 2002). Other groups that express dnTCFs also see p21 expression increase sharply - a finding not replicated here. This difference hints at a level of complexity above and beyond a simple model that TMED3 and SOX12 regulate direct WNT targets - and merits a measured, accurate conclusion statement in the Abstract and treatment in the Discussion.

We agree with the careful comments of this referee and hope to have provided measured and accurate conclusion statements. Context-dependency is a clear problem when working in cancer cells, unlike in normal development. This is why we have measured signatures with multiple genes. We never argued that these were core WNT signal transducers. For example, previous work on SOX proteins has documented their modifying role. No one would argue they are core components of WNT signaling but also no one would argue that they do not affect WNT-TCF outputs in specific contexts. We have revised the text accordingly.

3. TMED3 action: In response to reviewer requests the authors now provide a stronger piece of data to show that knockdown of TMED3 prevents secretion of a epitope-tagged Wnt ligand. This data strengthens their conclusions about TMED3 action.

We thank this reviewer for his/her comments and support.

4. That the authors provide an improved direct test with an epitope-tagged Wnt and have satisfied the concern from this reviewer is one thing, however this reviewer wishes to clarify a point made in the author's rebuttal. In response to the request by this reviewer that a specific inhibitor of Wnt ligand secretion be used instead of Brefeldin A to examine WNT ligand expression, the reviewers declined this suggestion stating that Porcupine inhibitors do not work in their hands, and that they are "not-so-clean pharmacological agents". It is important to clarify this point. IWP2 and C59 are quite specific small molecule inhibitors of Porcupine, an

ER-resident enzyme shown by multiple groups to function directly in Wnt signaling. Both inhibitors (and also the closely related LGK974 molecule) work at low nanomolar concentrations. There is also the next-generation IWP, IWP-L6 molecule and these operate at sub-nanomolar levels (Wang X, Moon J, Dodge ME, Pan X, Zhang L, Hanson J, et al. The Development of Highly Potent Inhibitors for Porcupine. J Med Chem. 2013). Brefeldin A blocks all protein transport from ER-to-Golgi and in so doing triggers ER stress and the Unfolded Protein Response (ultimately apoptosis). It is thus NOT valid to rely on a comparison of Brefeldin A and Wnt ligand localization and conclude that TMED3 promotes their secretion.

This referee agrees that we have satisfied her/his concern. We are thankful for this statement. We did not argue that Brefeldin A effects test for Wnt secretion. We use it as a test for Wnt mislocalization following Golgi perturbation, which we then compare with the phenotype of *shTMED3*. This has been clarified on p.14.

We now proceed to reply to his/her comments on Porcupine inhibitors: In our hands, neither C59 nor the much improved nanomolar range IWP-L6 molecule work well in human colon cancer cells. Following the very good suggestion of this referee we bought the new and improved Porcupine inhibitor. Treatments of colon cancer cells with these small molecules do not result in global changes in WNT-TCF response genes (A), and these do not mimic the responses to dnTCF (A). Moreover, the treatments do not affect WNT1 localization (B) and do not change WNT3A secretion (C) Figure (removed upon authors' request): A shows normalized ratios over controls. C59 and IWP L6 were used at 0.2 μ M (B) and 2 μ M (not shown, yielding the same negative results) for 12h; B left shows a confocal maximal projection stack, center and right single confocal planes; C shows a Western blot. See paper.

We do not intend to argue for or against Porcupine inhibitors as our paper do not address these molecules. We certainly hope they will work in the clinics but we note that the biological assays reported in the IWP-L6 paper (Wang et al., 2013) are exciting but perhaps not conclusive. IWP-L6 affects morphogenesis in ex vivo explant assays and zebrafish tails fail to re-grow fully. While suggestive, there are no genetic benchmarks or rescue assays. Additional tests include a Western blot testing phospho-Dishevelled levels with 2.5 μ M (!) treatments in HEK293 cells. Does it not work in colon cancer cells? Does it not work at lower concentrations since they claim action in the nanomolar range? We could not find any other papers. Similarly, for C59 we could find a single in vivo paper but using WNT-overexpression models (Proffitt et al., 2012). Finally, we have not been able to secure a source of LGK974 but we intend to test it in the future.

Our work is not on Porcupine inhibitors and we address this issue in response to the good comments of this referee. We regret not being able to be more positive on this point, but this is peripheral to our work and the present submission.

5. *Clonogenic Spheroid Assay for Stemness: Clonogenic survival in an anchorage-independent growth assay is certainly one way that aspects of stemness can be evaluated. But this reviewer does not see how the data in Fig. 6B is being misread. The percentage of each type of spheroid that develops under each condition is reported (both in the figure, panel B and in the legend). The text verbally restates the data shown in Fig. 6B. However, there is no place in the manuscript where the absolute number of spheroids that develop in each condition are given. Instead, the authors state that 15% of wildtype CC14 cells survive to develop spheroids. But then they then state on page 9 that:*

"...Cells with compromised SOX12 function showed a reduction by 50% in the number of large spheres but not of smaller ones (Fig. 6A,B). In contrast, shTMED3 induced a drastic ~10-fold decrease in large spheres with the majority (mostly small spheres) had a loose phenotype, with live single cells protruding from the clone and sometimes being separated from it (Fig. 6A,B). Both SOX12 and TMED3 thus appear required for efficient stem cell function but only the latter is also required for normal cell group compaction/adhesion...."

Stating the relative proportion of large versus small spheroids says nothing about stemness (altho this metric is interesting for what it might say about metastasis). This point is belabored here because the authors make a very strong conclusion about the role of TMED3 and SOX12 in both the Abstract and Discussion. What is needed is data showing the actual number of spheroids that develop in each condition.

We thank this referee for insisting on this issue. We hope to provide very clear descriptions so we welcome this opportunity.

As we understand his/her arguments, s/he does not like the quantification given as percentages. Multiple plates are used per experiment and the unit of clonogenic experimentation is the 'plate'. We have thus modified the y axis to state 'per plate'.

This referee also criticizes us for not explicitly providing the total number of spheroids per each condition. We already provided a quantification of the number of spheroids per class and per condition but to resolve this issue we now mention the total numbers of spheroids, explicitly, in the text: The total number of spheroids in triplicate experiments were 90 for control, 72 for *shSOX12* and 78 for *shTMED3*. Looking at these numbers, however, only does not give the main result, which is that only the 'large compact' spheroids are reduced in number for *shSOX12*. This is the message that the kind of graph we chose gives most clearly. And the same for *shTMED3*, except that in this case, the presence of a new phenotype, the 'loose' spheroids represent a second key point. We note that previous work has shown that only large spheroids are made by true clonogenic stem cells (versus committed progenitors) (Suslov et al., 2002) so that a reduction of large spheroids as in *shSOX12* can be interpreted as a reduction in one type of clonogenic stem cells.

To solve the problems with this section we have modified the y axis to clearly state 'Average number of clonal spheroids per 96 well plate' in Fig. 6. We have also nuanced the discussion regarding total numbers of clonogenic units, adding the Suslow et al reference on p.20. Moreover, as requested by this referee, we state the total number of spheroids formed in each case in the text on p.9, before describing the different phenotypes. We sincerely hope this referee will be satisfied and we thank him/her for giving us the opportunity to further clarify this section.

Referee #2 (Remarks):

In this entirely re-written manuscript, CC14 primary human colon cancer cells of the transit amplifying inflammatory type and HT29 colon cancer cells are used to demonstrate the role of the positive WNT pathway regulators TMED3 and SOX12 in suppressing metastases. The use of two types of colon cancer cells giving concordant results (Fig. 3, Fig. S9) and the revised focus significantly improve the manuscript.

We thank this referee for these positive notes.

Referee #3 (Comments on Novelty/Model System):

The authors use an innovative screening strategy for regulators of metastasis involving transduction of shRNA libraries followed by in vivo screening for lung metastasis of colon tumor cells. They identify two gene whose knockdown consistently increases metastasis of colon carcinoma cells. The experimental design is well described and experiments are neatly documented. The two candidates could have important functions in the metastatic progress and are therefore of medical interest. Moreover, a negative correlation of metastasis formation and activation of the Wnt pathway and a positive correlation to hedgehog signalling is indicated, in line with previous reports by the authors.

Referee #3 (Remarks):

Major critical points:

1 The enrichment of shRNAs in the primary screening approach towards genes that are not expressed in the injected tumor cells or that appear to act in opposite pathways (shTEAD) is a bit worrying. The authors ascribe these to "off-target effects" which would mean that their system is somehow sensitive to such effects. Although two different shRNAs to TMED3 and SOX12 support specificity the true control would be a rescue approach using transduction of shRNA by insensitive cDNA constructs.

We thank this referee for his/her comments. We note that all screens have false positive and false negatives. This is the reason why we carefully tested the candidates and why in the case of TEAD we tested also YAP. We think that there are several controls possible but the best control in our view is to obtain the same result in an independent manner, that is with a second totally independent shRNA

targeting the same mRNA but also in a different vector backbone. This is exactly what we have done. Our concern is not so much that one given shRNA is specific (and here the control suggested could be good indeed) but that the phenotype is specific as revealed by two independent shRNAs for the same gene in two independent vectors given that the chances of these having the same non-specific effects are basically nil.

2 Authors should be more cautious in the interpretation of their results. The fact that TMED3 and Sox12 can activate the Wnt pathway does not mean that their effects on metastasis are due to alterations of wnt signalling, although it would fit well with previous data from the group. Unless the authors show that activation of Wnt signalling, e.g. by active b-catenin would rescue effects of shTMED3 and shSOX12 the connection to Wnt signalling remains rather correlative. Their components could well have other functions in secretion of other factors (TMED3) or in gene expression of wnt unrelated genes (SOX12). This should be stated in the discussion and some of the passages extensively dealing with the Wnt connection might be shortened. It should also be noted that there was no effect of shTMED3 on TOP/FOP reporters which is difficult to reconcile with its effects on Wnt target gene expression.

We thank this referee for his/her word of caution. As requested we have performed rescue assays for the specificity of shTMED3 and shSOX12 for the WNT-TCF signature. We find that active β CATENIN rescues the expression of repressed TCF targets. This data is now incorporated in Fig. S9 both for CC14 cells and for HT29 cells. We hope this additional data will alleviate the concerns. Moreover, we have carefully revised the paper to give a measured assessment of the impact on WNT signaling.

Minor points:

Is there an explanation why shTMED3 and shSox12 cells outcompete the control cells in the red/green assay but show no difference to controls in tumor growth when grafted separately?

We can only suggest some options in the text. This is an issue under investigation.

The use of the term "primary colon cancer" to describe the CC14 cells is misleading as one would assume that tumor tissues have been grafted which was not the case. Instead one could replace this term by "primary colon tumor cells".

We thank this referee for highlighting this point. We have made sure we refer to 'primary colon cancer cells' and not 'primary colon cancers'.

What is the meaning of \pm 50% or \pm 60-90% used on pages 5 and 6?

These have been changed to 'about half' and '60-90%'.

*At certain points authors should be more precise:
p. 6 line 7 from bottom delete "than"?*

'Than' has been deleted. Thank you.

*p. 8 second paragraph: should probably read "The ability of the knockdown of ..."
Similarly, in the abstract: "knockdown of TMED3 ... induces full metastasis..."*

Corrected, thank you.

p14 third line from bottom: "However, it was only detected in the supernatant of control cells..."

Corrected, thank you.

p. 22 third line from bottom: It is useless to show formula if n, k are not described.

Corrected, thank you.

Thank you for the submission of your revised manuscript to EMBO Molecular Medicine. We have now received the enclosed reports from the Reviewers that were asked to re-assess it. As you will see the Reviewers are now globally supportive although there are a few remaining issues.

Briefly, Reviewer 1 would like you to provide the primary data for the spherogenesis experiments in the manuscript. Reviewer 3, instead, is not satisfied with your response to his/her request to verify whether active beta-catenin would rescue the effects of TMED and SOX12 KD on metastasis. Although I will not be asking you to provide further experimentation at this point, I would encourage you to provide the data if available, or in alternative to amend your text as to avoid overreaching conclusions. I am willing to make an Editorial decision on your final, revised version, provide the issues raised are dealt with as mentioned above.

Please also consider the following final Editorial amendments/requests:

1) Every published paper now includes a 'Synopsis' to further enhance discoverability. Synopses are displayed on the journal webpage and are freely accessible to all readers. They include a short standfirst - to be written by the editor - as well as 2-5 one-sentence bullet points that summarise the paper (to be written by the author). Please provide the short list of bullet points that summarise the key NEW findings. The bullet points should be designed to be complementary to the abstract - i.e. not repeat the same text. We encourage inclusion of key acronyms and quantitative information. Please use the passive voice. Please attach these in a separate file or send them by email, we will incorporate them accordingly

2) We are now encouraging the publication of source data, particularly for electrophoretic gels and blots, with the aim of making primary data more accessible and transparent to the reader. Would you be willing to provide a PDF file per figure that contains the original, un-cropped and unprocessed scans of all or at least the key gels used in the manuscript? The PDF files should be labelled with the appropriate figure/panel number, and should have molecular weight markers; further annotation may be useful but is not essential. The PDF files will be published online with the article as supplementary "Source Data" files. If you have any questions regarding this just contact me.

3) Upon submission of your revised manuscript, please remember to upload the improved image files as discussed during the quality control procedure.

I look forward to seeing a revised form of your manuscript soon, and possibly no later than two weeks.

***** Reviewer's comments *****

Referee #1 (Remarks):

Duquet and colleagues provide additional data and clarification in this revised manuscript. As stated before in the previous review, the screen developed by this team to look for metastasis regulators in the central strength of the study. Here, additional modifications to text in the Abstract and due diligence to questions/concerns from two of the three reviewers are provided. This manuscript encompasses a solid, well supported and annotated report. The overall screen and findings are important and will be of interest to a wide readership.

1. Wnt secretion

The authors have done due diligence with Porcupine inhibitors to address the Wnt secretion issue. They observe no effect on gene expression in colon cancer and no effect on Wnt secretion. Perhaps it is not surprising that there is little effect on gene expression in colon cancer cells because beta-catenin is already stabilized and the effects of autocrine Wnt ligand secretion do not matter - although at least one other group has recently reported active, autocrine Wnt activity in HT29 colon cancer cells and shown how they can block that activity with Porcupine inhibitors or Evi/Wls knockdown (Voloshanenko, O. et al. 2013. Nat. Comm.). Nevertheless, for the purposes of reporting on this new, interesting metastasis screen, this reviewer agrees that any further attention to this issue is beyond the scope of the study.

2. Tumor Spheroid forming assay

The authors have now provided the total number of spheroids that form under each condition to reviewers (90, 72 and 78 colonies from triplicate experiments). This primary data would also be useful to readers (e.g. a statement in the figure legend) as it helps assess the overall effect of TMED3 and SOX12 knockdown in the various assays. In its absence, readers might question whether there is a drastic difference in the numbers of colony formation from one condition to the other. As the author states however, there is not much of an effect on overall colony forming ability in the assay and that as he asserts, the main effect is on colony compaction and cell mobility for shTMED3 - a phenotype entirely consistent with the overall goal of the screen which is to search for metastasis regulators. Stating the numbers of spheroids that form also helps readers understand that the 2 knockdowns are not influencing survival, or some other aspect of clonogenic, stem cell activity, but appear to have a more focused effect on colony compaction and signaling.

Referee #3 (Remarks):

There were two major critical points in my previous review.

1. I tend to agree with the authors response to point #1 i.e. that the fact that two different shRNA produce similar effects is sufficient to prove specificity, although the proposed rescue experiments using shRNA insensitive cDNA constructs are meanwhile standard for certain journals.-
2. Maybe it was not so explicitly stated in the previous comment but the question was whether increased metastasis upon knockdown of TMED and SOX12 is indeed due to reduced Wnt signalling. It was proposed to determine whether active b-catenin would rescue these effects, i.e. reduce metastasis formation again. This experiment would certainly strengthen the functional link between reduced Wnt activity and increased metastasis upon loss of TMED and SOX12. Instead, what the authors did was to overexpress b-catenin and show that wnt dependent gene expression is restored which doesn't answer the above question.

Thank you for the submission of your revised manuscript to EMBO Molecular Medicine. We have now received the enclosed reports from the Reviewers that were asked to re-assess it. As you will see the Reviewers are now globally supportive although there are a few remaining issues.

Briefly, Reviewer 1 would like you to provide the primary data for the spherogenesis experiments in the manuscript.

We have now added the primary source data for the total number of spheroids in the figure legend, as requested.

Reviewer 3, instead, is not satisfied with your response to his/her request to verify whether active beta-catenin would rescue the effects of TMED and SOX12 KD on metastasis. Although I will not be asking you to provide further experimentation at this point, I would encourage you to provide the data if available, or in alternative to amend your text as to avoid overreaching conclusions.

We are happy that you do not require additional data. We have modified the text and trust it is now acceptable for EMBO Molecular Medicine. Below we provide a response to this referee's concerns.

I am willing to make an Editorial decision on your final, revised version, provide the issues raised are dealt with as mentioned above.

I look forward to seeing a revised form of your manuscript soon, and possibly no later than two weeks.

We look forward.

Referee #1 (Remarks):

Duquet and colleagues provide additional data and clarification in this revised manuscript. As stated before in the previous review, the screen developed by this team to look for metastasis regulators in the central strength of the study. Here, additional modifications to text in the Abstract and due diligence to questions/concerns from two of the three reviewers are provided. This manuscript encompasses a solid, well supported and annotated report. The overall screen and findings are important and will be of interest to a wide readership.

We thank this referee for his/her support.

1. Wnt secretion

The authors have done due diligence with Porcupine inhibitors to address the Wnt secretion issue. They observe no effect on gene expression in colon cancer and no effect on Wnt secretion. Perhaps it is not surprising that there is little effect on gene expression in colon cancer cells because beta-catenin is already stabilized and the effects of autocrine Wnt ligand secretion do not matter - although at least one other group has recently reported active, autocrine Wnt activity in HT29 colon cancer cells and shown how they can block that activity with Porcupine inhibitors or Evi/Wls knockdown (Voloshanenko, O. et al. 2013. Nat. Comm.). Nevertheless, for the purposes of reporting on this new, interesting metastasis screen, this reviewer agrees that any further attention to this issue is beyond the scope of the study.

We thank this referee for his/her comments and we thank him for his/her previous excellent suggestions.

2. Tumor Spheroid forming assay

The authors have now provided the total number of spheroids that form under each condition to reviewers (90, 72 and 78 colonies from triplicate experiments). This primary data would also be useful to readers (e.g. a statement in the figure legend) as it helps assess the overall effect of TMED3 and SOX12 knockdown in the various assays. In its absence, readers might question whether there is a drastic difference in the numbers of colony formation from one condition to the other. As the author states however, there is not much of an effect on overall colony forming ability in the assay and that as he asserts, the main effect is on colony compaction and cell mobility for shTMED3 - a phenotype entirely consistent with the overall goal of the screen which is to search for metastasis regulators. Stating the numbers of spheroids that form also helps readers understand that the 2 knockdowns are not influencing survival, or some other aspect of clonogenic, stem cell activity, but appear to have a more focused effect on colony compaction and signaling.

We note that the total number of spheroids was already added to the main text. We have now, in addition, added the single values of total spheroids for each condition for each of three independent experiments in the legend of Figure 6.

Referee #3 (Remarks):

There were two major critical points in my previous review.

1. I tend to agree with the authors response to point #1 i.e. that the fact that two different shRNA produce similar effects is sufficient to prove specificity, although the proposed rescue experiments using shRNA insensitive cDNA constructs are meanwhile standard for certain journals.-

We thank this referee for agreeing with us.

2. Maybe it was not so explicitly stated in the previous comment but the question was whether increased metastasis upon knockdown of TMED and SOX12 is indeed due to reduced Wnt signalling. It was proposed to determine whether active b-catenin would rescue these effects, i.e. reduce metastasis formation again. This experiment would certainly strengthen the functional link between reduced Wnt activity and increased metastasis upon loss of TMED and SOX12. Instead, what the authors did was to overexpress b-catenin and show that wnt dependent gene expression is restored which doesn't answer the above question.

We thank this referee for his/her views. We note, however, that the experiment s/he proposes is also an overexpression. We note also that we used activated b-CATENIN, as suggested, and that there is a rescue of TCF targets, which is fully consistent with our conclusions. Future studies will address additional issues, including in vivo analyses with other components.